# Metabolic reaction network-based recursive metabolite annotation for untargeted metabolomics

Xiaotao Shen[1,2], Ruohong Wang[1,2], Xin Xiong[1], Yandong Yin[1], Yuping Cai[1,2], Zaijun Ma[1,2], Nan Liu[1] & Zheng-Jiang Zhu [1]

Large-scale metabolite annotation is a challenge in liquid chromatogram-mass spectrometry (LC-MS)-based untargeted metabolomics. Here, we develop a metabolic reaction network (MRN)-based recursive algorithm (MetDNA) that expands metabolite annotations without the need for a comprehensive standard spectral library. MetDNA is based on the rationale that seed metabolites and their reaction-paired neighbors tend to share structural similarities resulting in similar MS2 spectra. MetDNA characterizes initial seed metabolites using a small library of MS2 spectra, and utilizes their experimental MS2 spectra as surrogate spectra to annotate their reaction-paired neighbor metabolites, which subsequently serve as the basis for recursive analysis. Using different LC-MS platforms, data acquisition methods, and biological samples, we showcase the utility and versatility of MetDNA and demonstrate that about 2000 metabolites can cumulatively be annotated from one experiment. Our results demonstrate that MetDNA substantially expands metabolite annotation, enabling quantitative assessment of metabolic pathways and facilitating integrative multi-omics analysis.

[1] Interdisciplinary Research Center on Biology and Chemistry, Shanghai Institute of Organic Chemistry, Chinese Academy of Sciences, 200032 Shanghai, P. R. China. [2] University of Chinese Academy of Sciences, 100049 Beijing, P. R. China. Correspondence and requests for materials should be addressed to Z.-J.Z. (email: jiangzhu@sioc.ac.cn)

Untargeted metabolomics aims to provide mechanistic insights by systematically characterizing metabolic changes in relevance to the physiological and pathological status at the system level[1,2]. Yet large scale and unambiguous annotation of metabolites remains a challenging task in liquid chromatogram-mass spectrometry (LC-MS)-based untargeted metabolomics[3,4]. Using accurate mass alone for metabolite annotation is impossible due to the high rates of false positive and high redundancy[5]. The widely used strategy in the identification of metabolites is to match experimental tandem mass spectrometry data (MS2 spectrum) with those from the standard spectral library[6,7]. This conventional method, however, is significantly limited by the fact that more than 90% of known metabolites in HMDB [http://www.hmdb.ca/] and METLIN [https://metlin.scripps.edu/] have no standard MS2 spectra[8], and that expanding the existing spectral library is hindered by the lack of chemical standards for many cellular metabolites. Moreover, this strategy suffers from having a poor standardization protocol for curating spectral libraries, due to uncharacterized spectral variations across different LC-MS instruments and laboratories[8]. Instead, significant efforts have been made to predict the MS2 spectra in silico for metabolites with high chemical diversity, such as MetFrag[9], CFM-ID[10], CSI:FingerID[11], MyCompoundID[12], MS-FINDER[13], and DEREPLICATOR+[14], and the accuracy for this approach has been substantially improved[15].

Alternatively, molecular and metabolic pathway can be utilized to facilitate the metabolite annotation. Several approaches, such as Mummichog[16] and PIUMet[17], map dysregulated metabolic features into a metabolic network and predict the pathway activity without unambiguous annotations of metabolites. However, these algorithms assume the local enrichment of dysregulated features into the specific pathways, which are sometimes inconsistently met or observed in biological samples[18]. Other approaches, such as Global Natural Products Social Molecular Networking (GNPS)[19], construct the molecular similarity network from the mass spectrometry data, and aim to annotate the unknown metabolites through annotated metabolite within the same sub-network. Specifically, Network Annotation Propagation (NAP)[20] and BioCAn[21] constructed the molecular network using molecular similarity and reaction information, respectively. Then, the metabolites in the network were matched to experimental and/or in silico predicted MS2 spectral databases. Then, it was hypothesized that one annotated node with more reliable neighbor metabolites was more accurate. So the annotations were re-ranked based on their neighbor metabolites. When the annotations are practically targeted to known chemical spaces in primary metabolisms, the information of metabolic reactions can be incorporated to increase the confidence of metabolite annotations.

Here, we develop a strategy, MetDNA (Metabolite annotation and Dysregulated Network Analysis), which implements a metabolic reaction network (MRN)-based recursive algorithm for metabolite annotation. In the cellular context, one metabolite can be catalyzed into another metabolite product. We thus define a reaction pair (or reactant pair, RP) by linking a substrate metabolite with its product metabolite displaying similar chemical structures. Notably in tandem MS, the fragmentation pattern of a metabolite is determined by its chemical structure; hence, two metabolites in a reaction pair tend to share similar MS2 spectra due to their structural similarities. As such, MetDNA can in principle annotate unknown metabolites using reaction-paired neighbor metabolites but not necessarily through existing MS2 spectra from the standard spectral library. More importantly, the reiterated application of this strategy allows progressive expansion of annotated metabolites via the recursive algorithm. Our data show that MetDNA significantly increases the number of annotated metabolites from less than 100 of conventional analysis to more than 2000 metabolites in one experiment. MetDNA also includes a self-check scoring system that diminishes the redundancy and uncertainty hits. Altogether, we propose that MetDNA allows a large-scale annotation of metabolites for untargeted metabolomics without a comprehensive standard spectral library, substantially advancing the omic-level study of metabolites for complex biological events.

## Results

**Reaction-paired metabolites share similar MS2 spectra**. We retrieved all RPs from the KEGG database (9603 RPs and 7639 metabolites) and further constructed a metabolic reaction network (MRN, Fig. 1a). In the MRN, one node represents one metabolite and two metabolites connected by an edge represent the reaction-paired neighbor metabolites. To test whether structurally similar metabolites in a reaction pair have a high chance to share similar MS2 spectra, we used three spectral databases of metabolites: our in-house library[22], NIST17 [https://chemdata.nist.gov/], and METLIN. While neighbor metabolites in RPs were retrieved, the same number of metabolites from non-RPs was also generated as controls. Then we compared four scoring methods (dot product (DP)[23], bonanza[24], hybrid similarity search (HSS)[25], and GNPS[26]) and finally chosen DP to compare MS2 spectral similarity between metabolites in RPs and non-RPs (Supplementary Note 1 and Methods). In our in-house library, more than 55.3% of reaction-paired neighbor metabolites have a DP score larger than 0.5. In contrast, only 5.2% of non-neighbor metabolites have a DP score larger than 0.5 (Fig. 1b). Consistently, the other two independent databases showed similar results (Fig. 1b). To demonstrate how the cosine similarities decay as a function of the reaction distance between two metabolites, we constructed RPs and non-RPs with two, three, four, and five steps, respectively. As shown in Supplementary Fig. 1, the percentages of RPs with DP > 0.5 significantly decreased to 26.5%, 15.5%, 10.3%, and 8.0% from 55.3% (with 1 step) for two, three, four, and five reaction steps, respectively. Combined, our data demonstrated that reaction-paired neighbor metabolites tend to share similar MS2 spectra compared to non-RP metabolites.

**MRN-based recursive metabolite annotation**. We developed the MetDNA workflow for the MRN based recursive metabolite annotation (Fig. 1c, d, Supplementary Fig. 2). To determine the effect, we applied the MetDNA workflow using untargeted metabolomics data from *Drosophila* aging samples (*Drosophila melanogaster*; 3 day vs. 30 day; Methods). For the positive mode dataset, we detected a total of 18,320 MS1 peaks (6428 have MS2 spectra) using XCMS[27]. We first annotated 134 metabolites using the standard spectral library. Among 134 metabolites, we selected 132 metabolites with KEGG IDs as the initial seed metabolites to map the MRN and retrieved their reaction-paired neighbor metabolites (Fig. 2a). At the first round, 654 neighbor metabolites were retrieved. Specifically, 150 of 654 metabolites were annotated by matching the calculated *m/z*, theoretical retention times (RTs) and surrogate MS2 spectra from the seed metabolites with the experimental data (Fig. 2a, Supplementary Figs. 3 and 4, details in Methods). However, 42 out of 150 metabolites were the same as seeds in the first round. Therefore, we excluded them as new seeds for next round of annotation to reduce the redundancy and computational cost in the recursive annotation. The seed selection was performed after each round of annotation. As results, we only selected 108 out of the 150 metabolites as new seed metabolites for the second round of annotation. Then, 529 reaction-paired neighbor metabolites were retrieved, and 299 were annotated using the surrogate

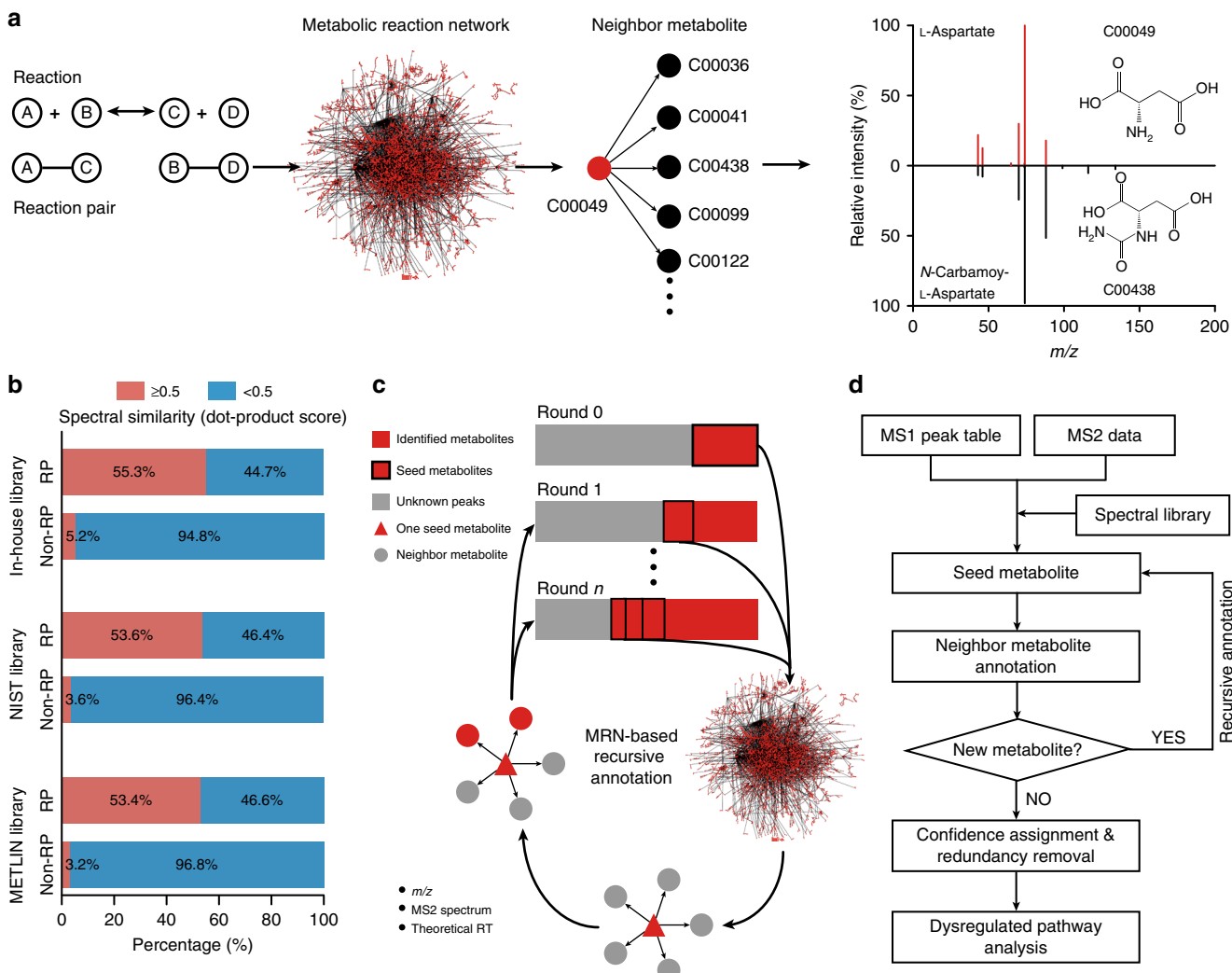

**Fig. 1** Metabolic reaction network and spectral similarity of neighbor metabolites in reaction pairs. **a** The construction of MRN using the reaction pairs retrieved from KEGG. An example is given to show that structurally similar reaction-paired neighbor metabolites have a high similarity of MS2 spectra. **b** Spectral similarity between metabolites in reaction and non-reaction pairs, respectively. Data were retrieved from three spectral libraries: in-house library, NIST17, and METLIN. **c** Illustration of MRN-based recursive annotation. Seed metabolites (red triangle) from round 0 are first mapped to MRN and all reaction-paired neighbor metabolites are retrieved (gray circle). The neighbor metabolites are then annotated by using the matching *m/z*, surrogate MS2 spectrum, and theoretical RT with unknown peaks (gray square, round 1). The recursive annotation runs until there are no new annotated metabolites (round *n*). **d** The overall workflow for MetDNA: (1) import of MS1 peak table and MS2 data; (2) annotation of initial seed metabolites; (3) annotation of reaction-paired neighbor metabolites; (4) MRN-based recursive annotation; (5) confidence assignment and redundancy removal; and (6) dysregulated pathway analysis

MS2 spectra from 108 seed metabolites. This seed selection-neighbor retrieval-neighbor annotation cycle was reiterated for 19 rounds, at which step no new metabolites could be annotated (Fig. 2a). We evaluated the confidence of the metabolite annotations and removed redundant hits (see Methods). For the positive mode, this analysis pinpointed a total of 1314 metabolites (Fig. 2a and Supplementary Figs. 3 and 4). We then repeated this analysis for the negative mode and determined a total of 1402 metabolites through 14 rounds of recursive annotation (Fig. 2b and Supplementary Figs. 5 and 6). In sum, a total of 1983 metabolites were cumulatively annotated from a single experiment (Fig. 2c). Notably, majority of the metabolites were resolved from the first eight recursive rounds, which accounted for more than 85% of total output. To improve the accuracy of the MetDNA workflow, the RT match threshold, weight for annotation scores, and score cutoffs were optimized according to annotation correct rate and propagation of misannotations

(Supplementary Note 2 and Supplementary Figs. 7–9). As a result, as the recursive annotation progressed, the confidence level for each round remained unchanged (Supplementary Fig. 10). According to definitions from Metabolomics Standards Initiative (MSI)[28], the annotation results given by MetDNA (except the initial seed metabolites) are level 3 of annotation.

To demonstrate an example, we conducted an in-depth assessment of L-arginine (KEGG ID: C00062), which, as a seed metabolite, resulted in the annotation of additional 28 metabolites by MetDNA (Fig. 2e). Among them, six metabolites were successfully validated using chemical standards, and six metabolites were further validated using NIST, METLIN, or HMDB library (Supplementary Fig. 11). The in silico MS2 spectra for the remaining 16 metabolites were predicted using CFM-ID[10], and compared to the experimental data (Supplementary Table 1). As a result, 6 out of 16 metabolites were validated with DP scores > 0.5, and 10 out of 16 metabolites had DP scores < 0.5. Similarly,

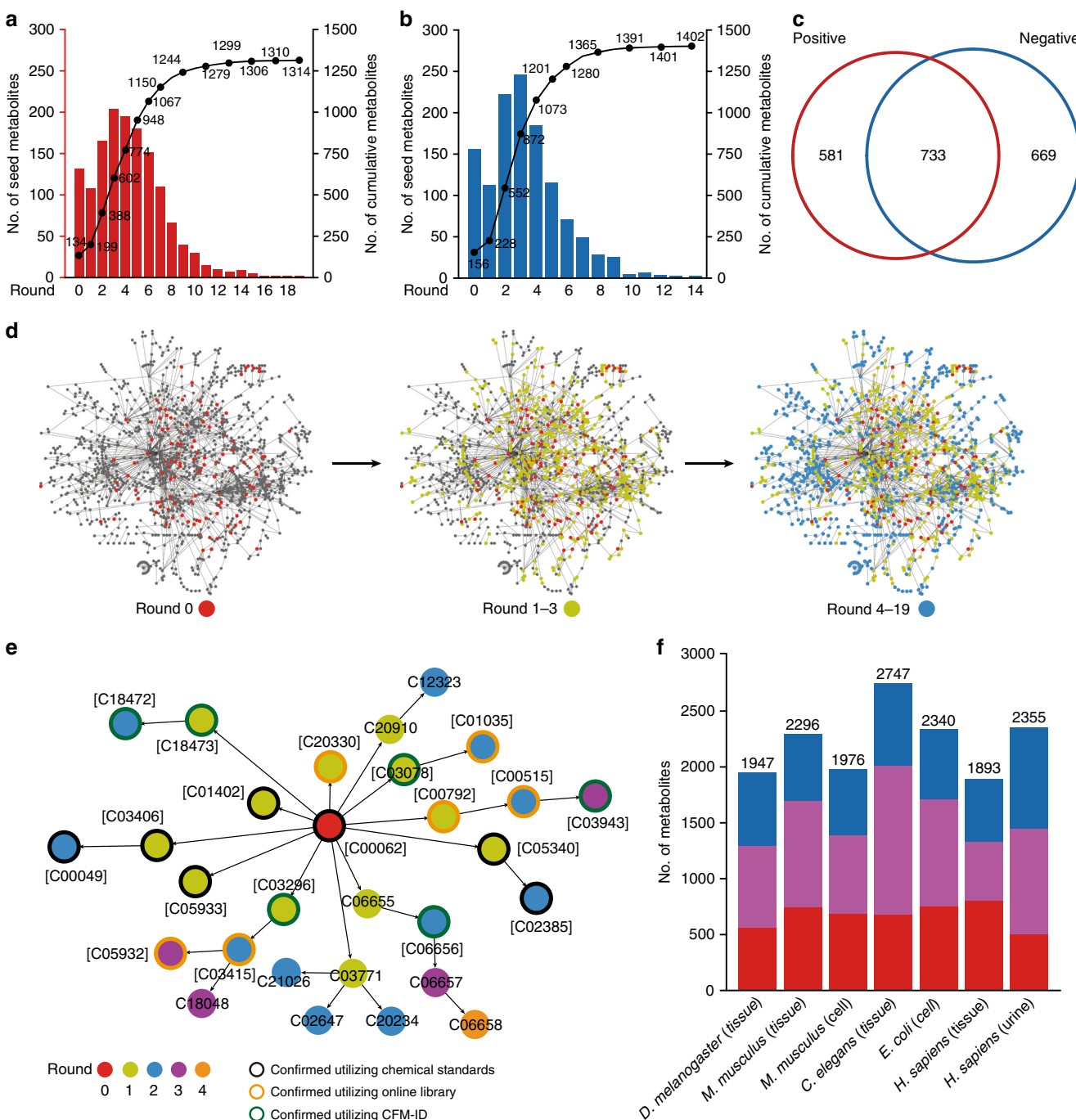

**Fig. 2** Metabolic reaction network (MRN)-based recursive metabolite annotation. The numbers of annotated metabolites in *Drosophila* aging datasets: **a** positive and **b** negative modes. The left *y*-axis of the plot is the number of seed metabolites in each round, and the right *y*-axis of the plot is the cumulative number of annotated metabolites. **c** A Venn diagram showing the overlap of annotated metabolites in positive (red) and negative modes (blue). **d** Network diagrams demonstrating the distributions of annotated metabolites in round 0 (left, red), rounds 1–3 (middle, yellow), and rounds 4–19 (right, blue). **e** An example of initial seed metabolite (L-arginine) is given to demonstrate how MetDNA identifies 28 reaction-paired neighbor metabolites. **f** The numbers of annotated metabolites in different biological species and samples. The red, blue, and purple bars represent the results from the positive mode, negative mode, and the overlap between the positive and negative modes, respectively

we also provided additional examples to illustrate the effectiveness of the recursive process (Supplementary Fig. 12).

We further examined MetDNA workflow with a variety of model organisms, including *Escherichia coli*, *Caenorhabditis elegans*, *Mus musculus*, and *Homo sapiens* with a wide array of sample types (prokaryotic cells, whole-body tissue, mammalian cells, brain tissue, liver tissue, colorectal tissue, and urine,

Supplementary Table 2 and Supplementary Note 3). As noted, data were acquired from three different instrument platforms (Sciex TripleTOF, Agilent QTOF and Thermo Orbitrap) and MS2 spectral data were obtained using different acquisition methods such as data-dependent acquisition (DDA), data independent acquisition (DIA), or targeted MS2 acquisition. Using MetDNA, our analysis consistently annotated more than

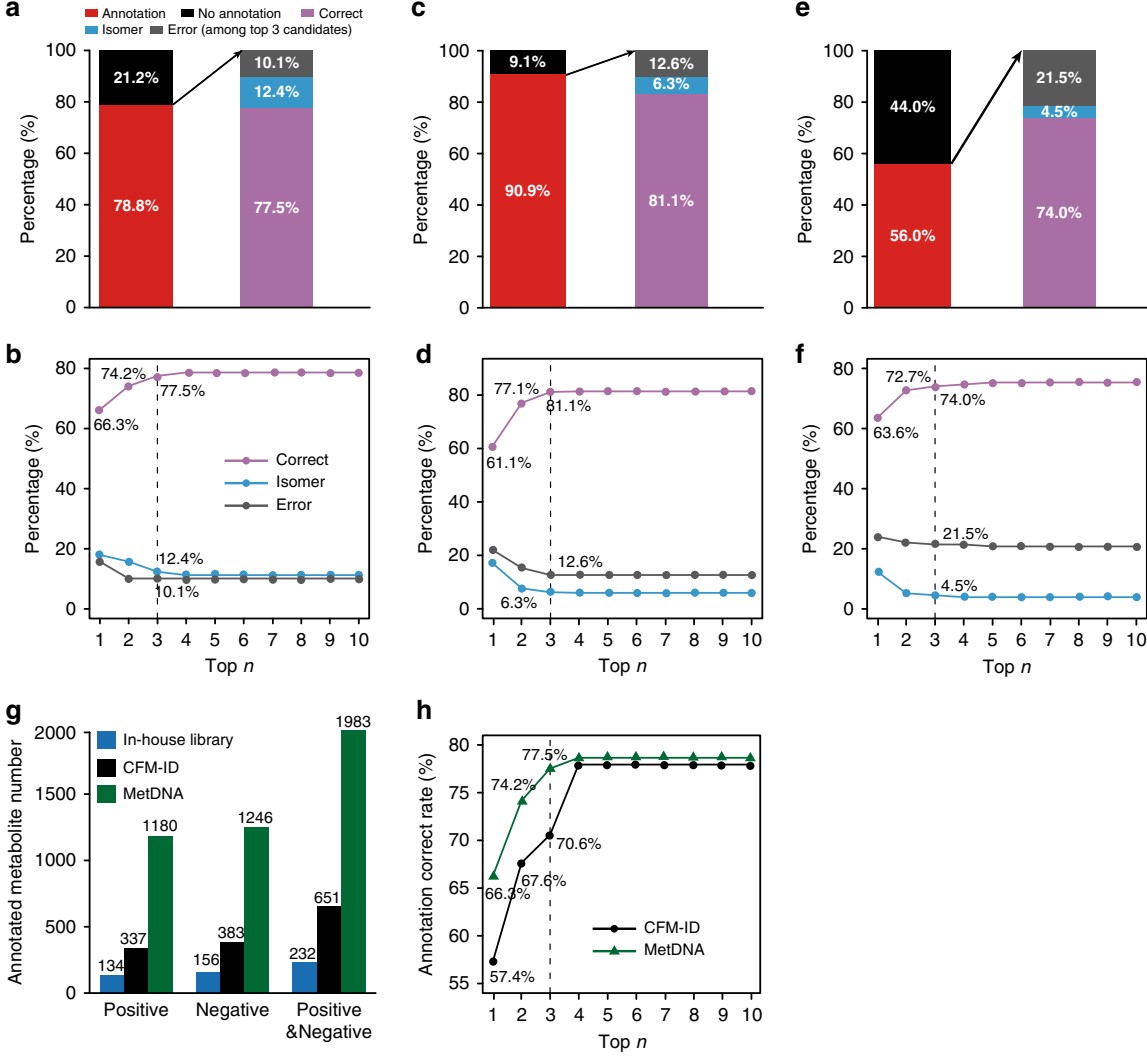

**Fig. 3** Validation of high annotation accuracy from MetDNA and the comparison with CFM-ID. **a**, **b** Validation of metabolite annotations from MetDNA by spiking 200 metabolite standards into mouse liver samples. **c**, **d** Validation of metabolite annotations using the confirmed seed metabolites in the *Drosophila* aging datasets. **e**, **f** Validation of metabolite annotations in *Drosophila* aging datasets from MetDNA using external NIST and MassBank spectral databases. **a**, **c**, **e** Statistics of the annotation coverage, and correct, isomer, and error rates among the top three candidates; **b**, **d**, **f** statistics of correct, isomer, and error rates among top $n$ ($n$ = 1 to 10) annotations. The $x$-axis represents the top $n$ ($n$ = 1–10) metabolite candidates for each peak. The $y$-axis represents the percentages for correct (purple), isomeric (blue), and erroneous (black) annotations. **g** The numbers of annotated metabolites using in-house library, CFM-ID, and MetDNA in *Drosophila* aging datasets. **h** Comparison of annotation correct rates between CFM-ID and MetDNA in the same chemical search space

2000 metabolites from single experiment (Fig. 2f, and Supplementary Table 3). Taken together, our data demonstrated that MetDNA is a platform-independent and versatile tool.

**Validation of MetDNA analysis accuracy.** MetDNA is a bioinformatic tool to annotate metabolites in real biological samples. Therefore, we designed the following validation experiments to evaluate the annotation accuracy of MetDNA (Fig. 3). We systematically compared the annotation results from the same biological samples between MetDNA and other approaches. Each validation experiment represented different confidence levels and different sizes of chemical search space. For the first experiment, a total of 200 (167 metabolites included in MRN) chemical standards were added into mouse liver samples as spike-in (Supplementary Data 1, Supplementary Note 4). Manual analysis of the experimental data with the comparison to the data of chemical standards ($m/z$, RT, and MS2 spectra) demonstrated that 113 and 137 metabolites were detected and identified in positive and

negative modes, respectively, which were considered as the level 1 identification according to MSI[28]. As a comparison, MetDNA successfully annotated 89 out of 113 (78.8%) and 113 out of 137 (82.5%) metabolites in positive and negative modes, respectively, indicating a high annotation coverage (Fig. 3a). All the MS2 spectra of 200 standards were intentionally removed from the standard spectral library, therefore, excluding them as the initial seed metabolites. The spike-in metabolites were all annotated through the MRN-based recursive strategy from round 2 to 9. Compared to chemical standards, we divided MetDNA annotation results into three categories: correct, isomeric, and erroneous annotations among top $n$ ranked candidates, which is similar to other metabolite annotation tools[10,11]. In our study, top three candidates were used to evaluate the performance of annotation except the specific statement. In MetDNA output for positive mode, the percentage of the correct annotation was 77.5% among top three candidates, whereas isomer and erroneous annotations were only 12.4% and 10.1% among top three

candidates, respectively (Fig. 3b, Supplementary Data 2). The average ranking of correct annotations was 1.3. Similar results were also obtained for negative mode of the mouse liver dataset (Supplementary Fig. 13, Supplementary Data 2).

For the second validation experiment, we first confirmed the structures of 107 seed metabolites using chemical standards in positive mode of the *Drosophila* aging dataset (Supplementary Data 3), which were also considered as the MSI level 1 identification[28]. We randomly chose 30% of the metabolites as seeds, and asked whether MetDNA could correctly annotate the rest 70% metabolites. This experiment has been repeated 10 times. As a result, a coverage of 90.9% of metabolites could be successfully annotated (Fig. 3c), which means that 90.9% of metabolites can be annotated by MetDNA. For the annotated metabolites, the percentages for the correct, isomeric, and erroneous annotations among top three candidates were 81.1%, 6.3%, and 12.6%, respectively (Fig. 3d). Similar results were also obtained for negative mode of *Drosophila* and *Escherichia coli* datasets (Supplementary Fig. 14 and Supplementary Data 4–6).

For the third validation experiment, we employed the external NIST17 and MassBank [http://www.massbank.jp] spectral databases (Supplementary Note 4). As noted, there are 2608 and 1464 metabolites in NIST and MassBank acquired on Q-TOF instruments, respectively. A total of 279 and 206 metabolites were identified from two spectral databases in positive and negative modes, respectively. The low ratio may be due to the insufficient measurement of metabolites in libraries on the particular LC-MS condition, or uncharacterized spectral variations across different LC-MS instruments and laboratories (i.e., between the experimental data and library data). Consistently, MetDNA successfully annotated 156 out of 279 (55.9%) and 128 out of 206 (62.1%) metabolites in positive and negative modes, respectively. The percentage for the correct annotation among top three candidates was 74.0%, whereas isomer and erroneous annotations were about 4.5% and 21.5%, respectively (Fig. 3e, f, Supplementary Fig. 15 and Supplementary Data 7). Collectively, these validation experiments demonstrated that the overall performance of MetDNA attains approximately 70–80% for correct annotations among top three candidates and 80–90% for correct formula annotation (correct and isomer) among top three candidates. In addition, the correct annotation rate among top three candidates should be considered as a true positive rate, which was 70–80% in different validation experiments. Meanwhile, the erroneous annotation among top three candidates rate should be considered as the false-positive rate, which was 10–20% in different validation experiments.

Finally, we also performed the benchmark comparison using in silico MS2 spectral prediction tool CFM-ID[10] (Supplementary Note 4). We predicted all the MS2 spectra of metabolites in MRN using CFM-ID with the pre-trained model and recommended parameters to construct an in silico MS2 spectra library, so the chemical space of this in silico MS2 spectral library is the same as MetDNA. It is worthy to note that although we and others[29] used the pre-trained model in CFM-ID to generate in silico MS2 spectra, the use of local dataset for training helps to improve the performance of these machine learning-based prediction tools. Then, we utilized the in silico MS2 spectra library for metabolite annotation of *Drosophila* aging datasets. As shown in Fig. 3g, 337 and 383 metabolites were annotated for positive and negative modes, respectively, and 651 metabolites were annotated in total. From the aspect of the annotation number, MetDNA provided a much higher number of annotation metabolites than CFM-ID with the same chemical search space (Fig. 3g).

We also used our first validation experiment to evaluate the annotation accuracy using CFM-ID. For the positive mode, the top 1–3 correct annotation rates for in silico MS2 spectral match

were 57.4%, 67.6%, and 70.6%, respectively. As a comparison, the correct annotation rates for MetDNA were 66.3%, 74.2%, and 77.5% respectively (Fig. 3h). The negative mode dataset had the similar results (Supplementary Figure 13). All these results demonstrated that MetDNA annotated much more metabolites and had higher annotation accuracy than in silico prediction tool CFM-ID. We further compared the predicted in silico MS2 spectra with experimental MS2 spectra acquired from the chemical standards (Supplementary Figure 13). The percentage of metabolites with DP scores larger than 0.5 is only 36.4%, and the median DP score was 0.33. These results demonstrated that the accuracy for the in silico MS2 spectra still requires further improvement.

**Feasibility of MetDNA for the small-sized spectral library**. To evaluate how the size of spectral library could impact the annotation result, we randomly selected a small fraction of seed metabolites from positive mode of the *Drosophila* aging dataset as the initial seeds (Supplementary Note 5). Interestingly, the use of only 21 metabolites as seeds was sufficient to annotate the similar number of metabolites compared to that of all 132 seed metabolites (Fig. 4a). In addition, about 88.2% of the annotated peaks had the exactly same annotations as those derived from all 132 seed metabolites (Fig. 5b). Similar result was obtained for the coverage of pathway enrichment analysis (Supplementary Fig. 16). However, when using more initial seed metabolites, the overall confidence levels were slightly elevated and consequently the redundancy in the final result was slightly reduced (Supplementary Fig. 17). We observed a similar result from metabolomics dataset of aging mouse samples (Supplementary Fig. 18). Therefore, the use of a small tandem spectral library is feasible for the MetDNA analysis, but an increase of tandem spectral library may help to improve the confidence level, while reducing redundant annotation of metabolites.

**Low propagation of misannotated metabolites**. A key feature of MetDNA is via recursive metabolite annotation. Misannotated metabolites may propagate during the reiterative process, therefore promoting false positive annotations. We constructed four possible types of misannotated metabolites compared to seed metabolites, including (1) metabolites with mass error > 25 ppm, and RT error > 60 s; (2) metabolites with mass error < 25 ppm, but RT error > 60 s; (3) metabolites with mass error < 25 ppm and RT error < 60 s; (4) metabolites with mass error < 25 ppm, but RT error > 60 s, and have high MS2 spectral similarity with seed metabolite (DP > 0.8, mostly isomers of seed metabolite, Supplementary Table 4). To gauge the consequence of misannotations, we intentionally replaced each of the 107 seed metabolites with four types of misannotated metabolites in the second validation experiment from positive mode of the *Drosophila* aging dataset, thus resulting four sets of seed cohorts that each represented one type of misannotated metabolites (Supplementary Note 6). For example, seed metabolite 4-aminobutanoate was replaced by phenylethylene (type 1; *m/z* error: 79.7 ppm; RT error: 234 s), diacetyl (type 2; *m/z* error: 0.7 ppm; RT error: 301 s), D-2-aminobutyrate (type 3; *m/z* error: 0.7 ppm; RT error: 36 s), or gamma-butyrolactone (type 4; *m/z* error: 2.5 ppm; RT error: 227 s; DP: 0.93).

Then MetDNA utilized these misannotated seed metabolites to perform MRN-based recursive analysis and compared with the control (i.e., using correct seeds, Fig. 4c). MetDNA analysis yielded 34, 143, 160, and 218 annotated metabolites, respectively, from above four sets of seed cohorts using type 1, 2, 3, and 4 misannotated metabolites. As a comparison, MetDNA yielded 1282 metabolites using the original correct seeds (Fig. 4d). The

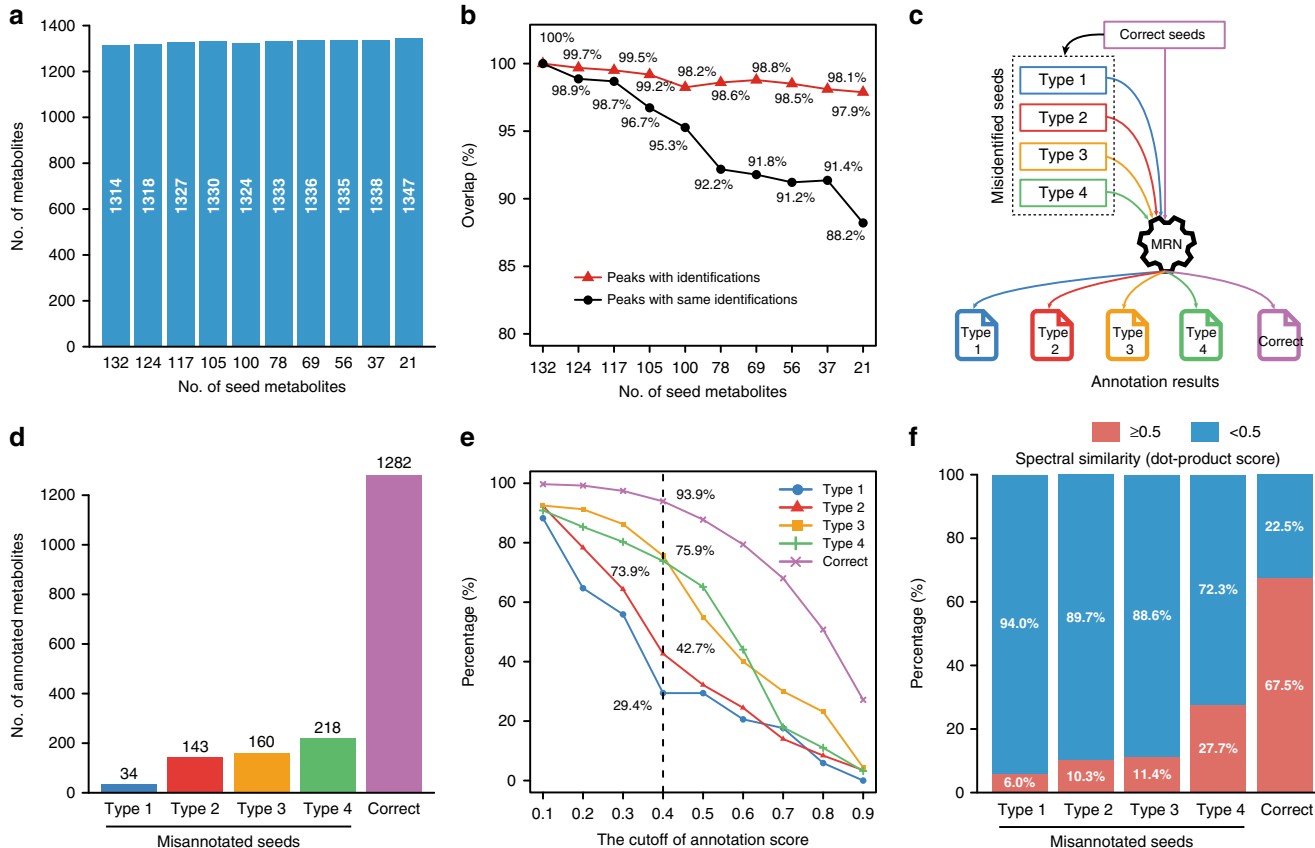

**Fig. 4** MetDNA is robust to small size library and has low propagation of misannotated metabolites. **a** The numbers of annotated metabolites using different numbers of initial seed metabolites. **b** The influence of the initial seed number on the annotations to mimic the different sizes of the tandem libraries. Red triangles represent the overlap percentages of MS1 peaks with annotations using different initial seed metabolites. Black circles represent the overlap percentages of MS1 peaks with the exact same annotations using different initial seed metabolites. In both cases, the annotation result using all 132 seed metabolites is used as a control. **c** The experimental design to evaluate the propagation of misannotated seed metabolites by replacing the correct seeds with four types of misannotated seeds. The types of misannotated seeds were defined as: Type 1: metabolites with mass errors were large than 25 ppm, and RT error were less than 60 s; Type 2: metabolites with mass errors were within 25 ppm, but RT error large than 60 s; Type 3: metabolites with mass errors and RT errors were within 25 ppm and 60 s, respectively; Type 4: metabolites with mass errors were within 25 ppm, but RT error were large than 60 s, and have high MS2 spectral similarity with seed metabolite (dot-product score was higher than 0.8). **d** The numbers of annotated metabolites using the correct and four types of misannotated seed metabolites. **e** The influence of the cutoff of annotation score on the numbers of annotated metabolites using the correct and four types of misannotated seed metabolites. **f** Spectral similarity between the correct and four types of misannotated seed metabolites and their reaction-paired neighbor metabolites

numbers of annotated metabolites were consistent with the degrees of error for misannotated metabolites. As compared from type 1 to type 4 of misannotated metabolites, they presumably represent a growing challenge in distinguishing, and therefore became more likely to propagate during the iterative process. However, even all seed metabolites were replaced as type 4 metabolites, the resulted error rate was only 17.0% (281 out of 1282), which was consistent with the validation experiments as shown above (i.e., 70–80% of correct annotations given by MetDNA). More importantly, by increasing the cutoffs of annotation scores, we observed that more annotations were removed from output results using misannotated seeds compared to controls (Fig. 4e). Therefore, an optimized cutoff score was set as 0.4, where 29.4%, 42.7%, 75.9%, and 73.9% of annotations were retained using types 1, 2, 3, and 4 of misannotated seed metabolites, but up to 93.9% of annotations were retained using correct seed metabolites (Fig. 4e, Supplementary Fig. 19). Finally, the mean number of edges (i.e., edge/node density) for correct seeds, type 1, type 2, type 3, and type 4 misannotated seeds (mean ± SEM) were calculated as 8.2 ± 2.8, 10.8 ± 2.9, 10.3 ± 3.9,

7.5 ± 3.6, and 6.1 ± 2.7, respectively. So the mean number of edges had no significant differences between correct seeds and four types of misannotated seeds. These results proved that the low propagation of misannotated seeds was not related to the edge density.

We further characterized the MS2 spectral similarity between the designed misannotated metabolites and their reaction-paired neighbor metabolites in MRN (Supplementary Note 6). Interestingly, from type 1 to type 4 misannotated metabolites, spectral similarity with their neighbors was considerably lower than that of control. The proportion of the spectral similarity larger than 0.5 was just 6.0%, 10.3%, 11.4%, and 27.7% for types 1, 2, 3, and 4 of misannotated metabolites, respectively. In a sharp contrast, this proportion rose to 67.5% for control seed metabolites. Combined, this analysis indicated that misannotated metabolites generally have low MS2 spectral similarity with their reaction-paired neighbor metabolites, and thus their propagation probability is intrinsically low within the MRN-based recursive analysis. Furthermore, the use of an optimized cutoff sore also helps to improve the annotation accuracy.

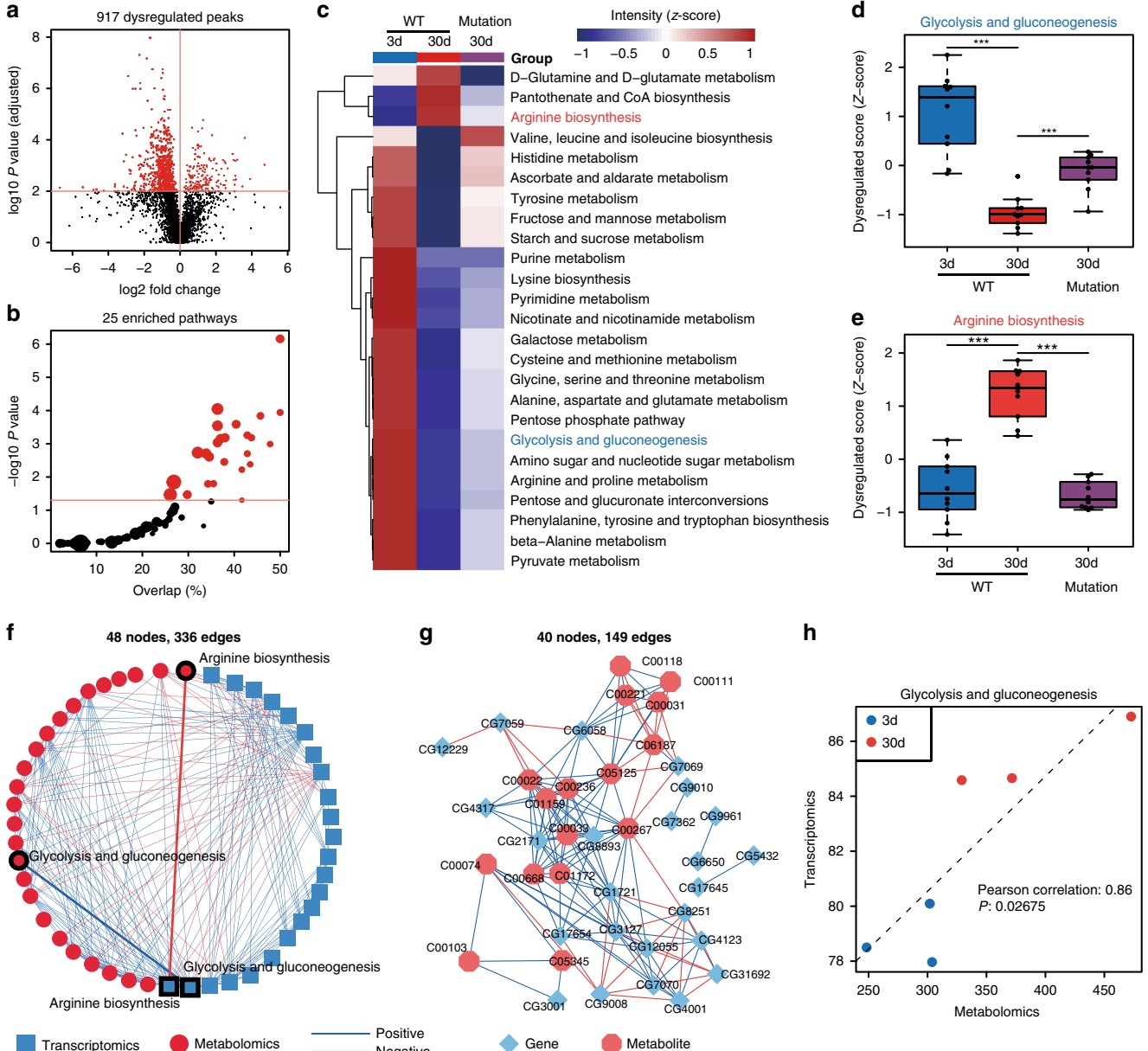

**Fig. 5** Dysregulated pathway analysis of aging fruit flies. **a** A volcano plot showing the dysregulated metabolites (red dots) in the *Drosophila* aging datasets (Student's *t*-test, FDR-corrected *P* values < 0.01). **b** The pathway enrichment analysis. The *x*-axis represents the overlap of annotated metabolites for each pathway. The *y*-axis represents the −log10(*P* value). The dot size represents the metabolite numbers in the pathway. The red color indicates the enriched pathways (hypergeometric test, *P* values < 0.05). **c** A heatmap showing the expression levels of 25 enriched metabolic pathways in aging fruit flies. The color indicates the *z*-score transformed intensity of the pathway. The mutant refers to PRC2 mutant. **d, e** Two box plots showing the expressions of the glycolysis and gluconeogenesis and arginine biosynthesis pathways, respectively. The upper, middle, and lower lines correspond to the first, second, and third quartiles (the 25th, 50th, and 75th percentiles). Student's *t*-test, *P* value < 0.001: \*\*\*; 0.001 < *P* value < 0.01: \*\*; 0.01 < *P* value < 0.05: \*; *P* value > 0.05: n.s. (no significant); *n* = 10 biologically independent samples for each group. **f** The correlative network between metabolic pathways in metabolomics and transcriptomics datasets. One node represents one metabolic pathway. The connections between two nodes were established by the Pearson correlation (Student's *t*-test, *P* values < 0.05). **g** The correlation network of genes and metabolites in the glycolysis and gluconeogenesis pathway. One node represents one gene or one metabolite. The connections were established by Pearson correlation (Student's *t*-test, *P* values < 0.05 and absolute Pearson correlation values > 0.7). **h** Correlation of the expressions for the glycolysis and gluconeogenesis pathway between transcriptomics and metabolomics

**Dysregulated metabolic pathways in aging flies**. With large-scale and accurate annotation, MetDNA can perform metabolic pathway analysis (Methods). During the *Drosophila* aging process, 917 dysregulated peaks (Student's *t*-test, FDR-corrected *P* values < 0.01, Fig. 5a) and 25 dysregulated metabolic pathways (hypergeometric test, *P* values < 0.05, Fig. 5b) were discovered. Most dysregulated pathways were associated with amino acid and

sugar metabolism (Fig. 5c). One prominent example was the glycolysis and gluconeogenesis pathway, which was decreased with age, but increased in PRC2 deficient animals, a long-lived mutant reported in our recent study[30] (Fig. 5d). In addition, MetDNA analysis revealed that the arginine biosynthesis pathway had a late-onset increase in old flies, but the extent of increase was diminished in age-matched PRC2 mutants (Fig. 5e). Since the

urea cycle included in arginine biosynthesis is the only metabolic route by which toxic ammonia can be converted into urea to be excreted from the body[31], these data suggested a negative correlation between arginine biosynthesis and urea cycle with adult lifespan.

Furthermore, a correlation network that combined metabolomics and transcriptomics datasets was constructed, which displayed a significant coherency between changes in metabolites and gene expression (Fig. 5f). The glycolysis and gluconeogenesis pathway, for example, was highly correlated at the level of metabolite and gene expression, as shown by a correlation network with 149 edges comprising 24 genes and 16 metabolites (Fig. 5g, h, Supplementary Fig. 20). Similar results were obtained for the arginine biosynthesis pathway (Supplementary Fig. 21). The high correlation between metabolites and gene expression further validated the high confidence of metabolite annotations using MetDNA.

## Discussion

In this work, we develop MetDNA, a MRN-based recursive strategy, to significantly increase the number of annotated metabolites without expanding the size of spectral library. MetDNA utilizes the principle that structurally similar metabolites in a reaction pair have a high chance to share similar MS2 spectra. However, we are aware that there are many occasions where a single metabolic biotransformation is sufficient to significantly alter the MS2 spectra between reaction-paired neighbor metabolites. MetDNA performs the recursive annotation only using spectrally similar neighbor metabolites. For example, although L-arginine in Fig. 2e has 26 neighbor metabolites through different reactions, MetDNA only recognizes reaction-paired neighbor metabolites having similar MS/MS spectra with L-arginine, such as beta-alanyl-L-arginine (DP: 0.96), N(omega)-hydroxyarginine (DP: 0.99) and N-(L-arginino)succinate (DP: 0.81). On the other hand, metabolites without similar MS/MS spectra with L-arginine cannot be annotated, such as agmatine (DP: 0.35), L-tyrosyl-L-arginine (DP: 0.23), and guanidine (DP: 0.28). Interestingly, the presence of L-arginine-derivatives without similar MS/MS spectra does not seem to affect the annotation of spectrally similar neighbor metabolites as well as recursive rounds. In addition, one node metabolite may have many neighbor metabolites in different metabolic reactions. If one of many neighbor metabolites has the similar MS2 spectrum as the node metabolite, it is enough to annotate this node metabolite. So even ~45% of the reaction-paired metabolites have DP score smaller than 0.5, the fact does not mean that all these metabolites could not be captured by MetDNA. It is worthy to note that low similarity between metabolites in non-RPs could effectively cease the propagation of wrong annotation in MRN-based recursive strategy (Fig. 4f). Collectively, high spectral similarity between reaction-paired neighbor metabolites ensures the high accuracy of annotation, while the reiterated application of this surrogate principle through our recursive algorithm allows significant and progressive expansion of annotated metabolites.

The comparison of spectral similarity between metabolites in RPs and non-RPs and the low propagation results of mis-annotated metabolites demonstrated that the false spectral similarity between metabolites in non-RPs, but not in RPs, causes the false positive annotations. Since the cosine similarities between metabolites in non-RPs remained around 5.0% (Supplementary Fig. 1), we reason that the false-positive annotation is not related to the reaction steps. Meanwhile, the spectral similarity between metabolites in RPs determined the capability and coverage to annotate the neighbor metabolites within one or multiple reaction steps. Clearly, the cosine similarities significantly decay along the

increasing reaction steps between metabolites in RPs. This evidence implied that the challenge for annotation of metabolites becomes increased with multiple reaction steps. However, if two reaction-paired metabolites are spectrally similar, it can be annotated by MetDNA regardless of their reaction steps (either one or multiple). In MetDNA, we set up to three reaction steps for the search for neighbor metabolites.

Recently, some publications utilized the molecular or reaction network for metabolite annotation, such as GNPS-NAP[20] and BioCAn[21]. The core concept of GNPS-NAP tool was to connect experimental metabolic features in a molecular network according to their MS2 spectral similarity. Then, the authors hypothesized that the neighbor features with similar MS2 spectra tended to share the similar chemical structures. This hypothesis was used to improve the annotation accuracy of peaks which were first annotated by in silico predicted MS2 spectra. In principle, the NAP work employed the spectral similarity of neighbor features (instead of reaction-paired neighbor metabolites in MetDNA) to re-rank the annotations obtained from in silico predicted MS2 spectra. Finally, no recursive annotation was employed in NAP which limited the effective coverage. For BioCAn, it first constructed a network using reaction information. Then, the metabolites in the network were mapped to the MS1 peak table using the m/z match, and generated a small size network. Then, the experimental MS2 spectra for matched features were further matched to different MS2 spectral databases and in silico MS2 prediction tools to calculate node score. Then, the authors hypothesized that one annotated node with more reliable neighbor metabolites is more accurate. So the annotation score was calculated to sum all node scores from the metabolites and their neighbor metabolites. Similar to the GNPS-NAP, BioCAn intended to re-rank the annotations obtained from the MS2 spectral databases and in silico MS2 prediction tools. However, no spectral similarity between reaction-paired neighbor metabolites was utilized. Similar to GNPS-NAP, there was also no recursive annotation in BioCAn. Therefore, we think the concepts of NAP and BioCAn are very different from our MetDNA. In addition, the reason why GNPS does not use the metabolic reaction network is that it focuses on natural products and is highlighted by annotating new chemical structures. MetDNA, however, mainly focuses on the annotation of known metabolites for primary metabolisms. It is well known that KEGG database includes very limited numbers of natural products and lipids. Thus MetDNA is not applicable for natural product annotation or untargeted lipidomics where the metabolic pathways and reactions are not well defined.

In MetDNA, the KEGG is used to construct MRN, because KEGG is the most important and popular database in biology, and is one of the most curated databases for metabolomics. In KEGG, the substrate–product metabolite pairs in a metabolic reaction were manually curated and validated for each reaction, which makes them highly reliable[32]. In this work, we chose the KEGG database for MetDNA due to its reliable definition of reaction pairs and neighbor metabolites[32]. A total of 6439 metabolites from KEGG that were defined in reaction pairs were retrieved for MRN, which is a relatively small number compared to other databases. But we think it is large enough to demonstrate the improved identification accuracy provided by MetDNA. For example, the 917 dysregulated peaks in *Drosophila* aging datasets were chosen for annotation using the metabolites in MRN. As a result, the average number of annotation given by MetDNA for 917 dysregulated peaks is only 1.6. This number means that one peak has only 1.6 metabolite candidates on average. As a comparison, the annotation redundancy is 16.1 when only using m/z and predicted RT match (the redundancy is 34.4 when only using m/z match, see Supplementary Note 8 and Supplementary

Data 8). Taken together, these results demonstrated that MetDNA significantly increases the number of annotated metabolites, while effectively reducing the redundancy thereby improving the accuracy of annotation. The high correct rate of annotation using MetDNA can be demonstrated even with a relatively small chemical space such as KEGG. In the future, other expanded biochemical reaction databases such as SMPDB[33], MetaCyc[34], and Reactome[35] could also be considered to further expand the coverage of MetDNA. For example, Blazenovic et al.[29] have recently reported to annotated all MS/MS spectra obtained from urine samples in untargeted metabolomics through multiple methods and very large chemical search spaces including 1102 authentic standards, MoNA and NIST17 libraries (13,808 compounds), CarniBlast library (2400 acylcarnitine species), CSI: FingerID (a filtered version of Pubchem containing 270,000 structures), and NIST17 hybrid search. Using CSI:FingerID alone, the authors demonstrated that 728 out of 6447 MS/MS spectra (11.3%) were assigned to chemical structures. Therefore, although about 2000 metabolites can cumulatively be annotated from one experiment with a relatively small chemical search space in MetDNA, expanding the chemical spaces in MetDNA and combined with other tools in the future could effectively increase the number of annotated metabolites.

We compared four different scoring methods to evaluate MS2 spectral similarity between metabolites in RPs and non-RPs, namely, DP[23], bonanza[24], HSS[25], and GNPS[26] scores (Supplementary Note 1). The results indicated that the use of GNPS and HSS scores will facilitate to increase the annotated metabolite numbers in MetDNA. However, it may also subsequently increase the false positives given the fact that the spectral similarity scores between metabolites in non-RPs become increased (5.2% using DP vs.10.4% using GNPS and 10.2% using HSS, see Supplementary Fig. 1). On the contrary, the use of bonanza score may have an inverse effect (i.e., lower annotation numbers and false-positive rate, Supplementary Fig. 1). For MetDNA, we think the DP score is suitable for connecting the nodes defined in MRN. Other scores such as GNPS and HSS should be very useful to annotate new chemical structures for natural products and lipids. To incorporate the other scores into MetDNA, the systematic optimizations of score cut-off and evaluation of the validation results must be required.

Finally, metabolomics data acquired using mass spectrometry are very challenging to result in an absolutely unambiguous structure. Current MS instruments cannot differentiate stereoisomers or highly similar isomers (such as adenosine 5'-monophosphate vs. adenosine-3'-monophosphate) since they share the same $m/z$, RT, and MS2 spectra. As a consequence, one peak is commonly annotated with several possible metabolite candidates in untargeted metabolomics (or called annotation redundancy). MetDNA outputs the top five ranked annotation candidates with match scores higher than 0.4, which is similar to other common tools (e.g., XCMS-Online [https://xcmsonline.scripps.edu/], and Sirius [https://bio.informatik.uni-jena.de/software/sirius/]). In MetDNA, each annotation is also assigned with a level of confidence, and redundant annotations are removed using the level of confidence through a multi-step and recursive strategy. However, MetDNA cannot exceed the limit of mass spectrometry or replace the chemical standards that ultimately confirm the unambiguous annotation of metabolites[28].

## Methods

**Metabolic reaction network**. The metabolic reaction network is a network for metabolite-to-metabolite-based enzymatic reactions, and it was constructed using the KEGG database. The KEGG compound database was downloaded using an R package KEGGREST in Bioconductor on 7 March 2017. We have provided this version of KEGG compound database as an R file named as kegg.compound in

Supplementary Data 9. In one metabolic reaction, the substrate and product metabolites are paired according to their structural similarity, and defined as one reaction pair (or reactant pair, RP). The KEGG reaction pair database was originally defined as RPAIR in KEGG[32]. This database was also obtained using the R package KEGGREST on 7 March 2017. We have also provided this version of KEGG reaction pair database as an R file named as kegg.rpair in Supplementary Data 10. We did not filter any reaction pairs from the KEGG reaction pair database, and all of reaction pairs were directly imported into R package igraph [https://igraph.org/r/] to construct the MRN. In the MRN, one node represents one metabolite, and one edge represents one reaction pair. Two metabolites connected by one edge indicate that they are reaction-paired neighbor metabolites. Finally, the MRN contains 7639 formulas (nodes) and 9603 reaction pairs (edges) in total. However, 1200 out of 7639 formulas were not organic molecules or the metabolites with a strict chemical structure, so the chemical space for MetDNA is 6439 metabolites in total. Detailed information of all the reaction pairs in the MRN is provided in Supplementary Data 11 and Supplementary Data 10.

**Data import**. MetDNA requires the import of an MS1 peak table (.csv format) and MS2 data files (.mgf or .msp format). The MS1 peak table is a list of metabolic peaks with annotated $m/z$ and RTs. The MS1 peak table is generated from the raw MS files using common peak picking software such as XCMS[27] and MS-DIAL[36]. MS2 data files (.mgf format) are converted from MS raw files using ProteoWizard [http://proteowizard.sourceforge.net/] (version 3.0.6150). MS2 data from different data acquisition methods such as DDA, DIA, or targeted MS2 acquisition are all supported. If MS-DIAL is used for peak picking in DIA data, the generated MS2 data files (.msp format) are also supported by MetDNA. The details and examples about the generation of the MS1 peak table and MS2 data files are provided in Supplementary Note 7.

**Annotation of initial seed metabolites**. With the imported MS1 peak table and MS2 data, MetDNA first matches the MS1 peaks with MS2 spectra according to their $m/z$ (±25 ppm) and RT (±10 s) values. If one MS1 peak matches multiple MS2 spectra, the most abundant MS2 spectrum is selected. In MetDNA, the intensities of the top 10 abundant fragment ions are summed up to represent the abundance of MS2 spectra. If one uses MS-DIAL for the peak picking and outputting MS2 data (.msp format), then the MS1–MS2 match step is skipped by MetDNA. The generated MS1/MS2 pairs are matched with our in-house standard spectral library for metabolite annotation. The match tolerance for the MS1 $m/z$ value is set as ±25 ppm. We compared four different scoring methods to evaluate MS2 spectra similarity (see Supplementary Note 1), and found that DP score is an appropriate one. So the modified DP function[23] is used to score the similarity between the experimental spectrum and the standard spectrum in the library (Eq. 1). The DP score ranges from 0 to 1, from no match to a perfect match. The intensities of the fragment ions in the MS2 spectra are rescaled so that the highest fragment ion is set to 1.

$$\text{Dot product} = \frac{\sum W_S W_E}{\sqrt{\sum W_S^2 W_E^2}}, \qquad (1)$$

where weighted intensity vector, $W = [\text{relative intensity of fragment ion}]^n [m/z \text{ value}]^m$, $n = 1$, $m = 0$; S = standard, and E = experiment. DP scores from both forward and reverse matches are generated. Metabolite annotations with DP scores in either forward or reverse matches larger than 0.8 are kept. Usually, 50–200 metabolites are annotated using the standard spectral library. The annotations are further filtered using the theoretical RTs of the metabolites (±30%). The generation of the theoretical RTs is provided below. The remaining annotations are used as initial seed metabolites in round 0 for MRN-based recursive annotation. The seed metabolites are defined as the annotated metabolites that provide their MS2 spectra as surrogate MS2 spectra for the annotation of their reaction-paired neighbor metabolites.

**Annotation of isotope and adduct peaks**. First, the isotope peaks of seed metabolites are annotated by MetDNA (Supplementary Fig. 2). For each seed metabolite, the program calculates the theoretical $m/z$ and the relative intensities of isotope peaks from the formula using the binomial and McLaurin expansion (R package Rdisop). Only four isotope peaks ([M] to [M+4]) are calculated. The generated isotope peaks of seed metabolites are used to match all the MS1 peaks in the MS1 peak table according to the $m/z$, RT and relative intensity. The default tolerances for the $m/z$, RT, and relative intensity are set as ±25 ppm, ±3 s, and ±500%, respectively. The large tolerance for the relative intensity match is due to the inaccuracy of the experimental intensity ratios for the isotope peaks[37,38].

The matched isotope peaks are assigned the annotation scores (Score$_{\text{iso}}$) shown below:

$$\text{Score}_{\text{iso}} = \text{Score}_{m/z} W_{m/z} + \text{Score}_{\text{RT}} W_{\text{RT}} + \text{Score}_{\text{int}} W_{\text{int}}, \qquad (2)$$

where Score$_{m/z}$ represents the $m/z$ match score and is calculated as follows:

$$\text{Score}_{m/z} = 1 - \frac{(|(mz_E - mz_T) \times 10^6 / mz_T|)}{\text{Tolerance}_{m/z}}, \qquad (3)$$

$mz_E$ and $mz_T$ are the experimental $m/z$ and theoretical $m/z$, respectively. The Tolerance$_{m/z}$ represents the tolerance for the $m/z$ match with a default value of 25 ppm.

Score$_{RT}$ represents the RT match score and is calculated as follows:

$$\text{Score}_{RT} = 1 - \frac{(|RT_E - RT_T|)}{\text{Tolerance}_{RT}}, \qquad (4)$$

where $RT_E$ and $RT_T$ are the experimental RT and theoretical RT, respectively. The Tolerance$_{RT}$ represents the tolerance of RT match with a default value of 3 seconds.

Score$_{int}$ represents the relative intensity match score and is calculated as follows:

$$\text{Score}_{int} = 1 - \frac{(|(Int_E - Int_T) \times 100/Int_T|)}{\text{Tolerance}_{int}}, \qquad (5)$$

$Int_E$ and $Int_T$ are the experimental relative intensity and theoretical relative intensity, respectively. The Tolerance$_{int}$ represents the relative intensity match tolerance with a default value of 500%. $W_{m/z}$, $W_{RT}$, and $W_{int}$ represent the weights of Score$_{m/z}$, Score$_{RT}$, and Score$_{int}$, respectively, and the default values are 0.45, 0.45, and 0.1, respectively.

Second, MetDNA calculates the possible $m/z$ values for the adduct peaks of seed metabolites. The adduct table for different LC separations (HILIC or RP) and polarities (positive and negative) are listed in Supplementary Table 5. All the possible adduct peaks of the seed metabolites are then matched to the MS1 peak table according to their $m/z$ and RT. The default tolerances for the $m/z$ and RT are ± 25 ppm and ±3 s, respectively. The annotations of the adduct peaks are also assigned annotation scores (Score$_{adduct}$) as follows:

$$\text{Score}_{adduct} = \text{Score}_{m/z} W_{m/z} + \text{Score}_{RT} W_{RT}, \qquad (6)$$

where Score$_{m/z}$ represents the $m/z$ match score and is calculated as indicated in Eq. (3). Score$_{RT}$ represents the RT match score and is calculated using Eq. (4). $W_{m/z}$ and $W_{RT}$ represent the weights of Score$_{m/z}$ and Score$_{RT}$, respectively, and the default values are 0.8 and 0.2, respectively.

Third, after the annotation of the adduct peak, the isotope peak annotation for each adduct peak is also performed using the same procedures described above.

The annotation of in-source fragmentation is not included in MetDNA. But some in-source fragmentations can be well differentiated using the RT. For example, RTs for tryptamine and tryptophan are 325.1 and 446.7 s, respectively, which generate an RT error of 37.2%. In MetDNA, true tryptamine and tryptamine from the in-source fragmentation can be well differentiated using the RTs.

**Annotation of reaction-paired neighbor metabolites**. Two metabolites in one reaction pair are neighbor metabolites. The seed metabolites provide their experimental MS2 spectra as surrogate spectra to annotate their neighbor metabolites. First, the initial seed metabolites are mapped to MRN and retrieve their neighbor metabolites from MRN. The reaction step is defined as the number of reactions between two metabolites. All the neighbor metabolites with one reaction step are retrieved. The MS2 spectra of the seed metabolites are assigned to their corresponding neighbor metabolites as surrogate MS2 spectra. The precursor $m/z$ was calculated for the neighbor metabolites according to the possible adduct (such as $[M+H]^+$ or $[M+Na]^+$) in the LC condition (HILIC or RP) and ionization polarity. In addition, RTs for the neighbor metabolites were also predicted. For each neighbor metabolite, the generated theoretical $m/z$, predicted RT (from RT prediction), and surrogate MS2 spectrum are matched to the experimental MS1 $m/z$ (default tolerance: ±25 ppm), RT (default tolerance: ±30%), and MS2 spectrum (default tolerance: 0.5). If one seed metabolite leads to no annotation of a neighbor metabolite with one reaction step, then neighbor metabolites with two reaction steps are further retrieved. The default value of the maximum reaction step is 3. The annotations of neighbor metabolites are assigned the annotation scores shown below:

$$\text{Score}_{iden} = \text{Score}_{m/z} W_{m/z} + \text{Score}_{RT} W_{RT} + \text{Score}_{spec} W_{spec}, \qquad (7)$$

where Score$_{m/z}$, $W_{m/z}$, and $W_{RT}$ are the same as in Eq. (2). Score$_{RT}$ is calculated as follows:

$$\text{Score}_{RT} = 1 - \frac{(|(RT_E - RT_T) \times 100/RT_T|)}{\text{Tolerance}_{RT}}, \qquad (8)$$

where $RT_E$ and $RT_T$ are the experimental RT and theoretical RT, respectively. The Tolerance$_{RT}$ represents the tolerance of RT match with a default value of 30%.

Score$_{spec}$ is the MS2 spectral match score, which is scored using a dot-product function (Eq. (1)) with some modifications. If the $m/z$ value of seed metabolite is larger than the neighbor metabolite, the fragment ions in the surrogate MS2 spectrum with $m/z$ larger than that of the neighbor metabolite are removed. Vice versa, the fragment ions in the experimental MS2 spectrum with $m/z$ larger than that of seed metabolite are also removed. $W_{spec}$ is the weight of Score$_{spec}$. The default values of $W_{m/z}$, $W_{RT}$, and $W_{spec}$ are 0.25, 0.25, and 0.5, respectively. The annotations of each MS1 peak are ranked by Score$_{iden}$. After the annotation of neighbor metabolites, isotope peak annotation is also performed for each neighbor metabolite using the same procedures described above.

**Selection of seed metabolites and recursive annotation**. After one round of annotation, new seed metabolites are selected from the annotated neighbor metabolites to start recursive annotation (Fig. 1c). Peaks with new annotations are selected as seed metabolites. MetDNA then repeats the neighbor metabolite identification and isotope peak annotation until there are no new seed metabolites available for the next round of annotation.

**Confidence assignment**. MetDNA uses a multi-step strategy to evaluate the confidence of the metabolite annotation. First, all the annotated MS1 peaks are grouped according to their annotation and RT. The MS1 peaks with the same metabolite annotations are grouped together and further divided into different peak groups according to their RT. The peak group is defined as a set of peaks (e.g., monoisotope peak, isotope peaks, and adduct peaks) with the same annotation and in the same RT window (default is 3 s). If one peak has multiple annotations, it may belong to multiple peak groups. Similarly, one metabolite may also belong to multiple peak groups. The confidence is then assigned to each peak group and all MS1 peaks in the group according to the following rules:

(1) Grade 1: at least one MS1 peak in the peak group is annotated through the standard spectral library (or initial seed metabolite);
(2) Grade 2: do not meet rule 1, and there are isotope peaks available in the peak group;
(3) Grade 3: do not meet rules 1 and 2, and there are reliable adduct peaks in the peak group, such as $[M+H]^+$, $[M+Na]^+$ or $[M+NH_4]^+$ for positive mode, and $[M-H]^-$, $[M+Cl]^-$, or $[M+CH_3COO]^-$ for negative mode, respectively; and
(4) Grade 4: the remaining peak groups that do not meet rules 1, 2, or 3.

**Redundancy removal**. Annotation redundancy includes peak redundancy and metabolite redundancy. Peak redundancy is defined as the total number of metabolite annotations divided by the total number of peaks with annotations, that is, the number of metabolites per peak. By contrast, the metabolite redundancy is defined as the total number of peak groups with annotation divided by the total number of metabolite annotations, that is, the number of peak groups per metabolite. MetDNA then removes the annotation redundancy according to the confidence levels of the peak group and the peaks in the group. First, if one metabolite matches multiple peak groups, the program removes the annotation from all the peaks in the peak groups with grade 4. However, if all of the matched peak groups are grade 4, all the annotations are maintained. Second, if one peak matches multiple metabolites, the annotations with the highest grade are kept. After the removal of the annotation redundancy, the constitution of the peak groups may change. The confidence assignment is then repeated, followed by a repetition of the redundancy removal process, which is also a recursive process. The recursive process continues until the annotation redundancy remains unchanged. The annotation redundancy is calculated as the mean value of the peak redundancy and the metabolite redundancy.

**Identification of dysregulated pathways**. A pathway enrichment analysis is used to identify and characterize the dysregulated metabolic pathways. First, dysregulated peaks with annotations are selected according to a univariate test (Student's $t$-test or Mann–Whitney–Wilcoxon test) with or without FDR correction. The maximum tolerance of $P$ values can be set by the users. A volcano plot is provided to demonstrate the distributions of the dysregulated peaks. Second, the metabolite annotations from the dysregulated peaks are mapped to the KEGG metabolic pathways. Currently, MetDNA contains the pathway information for 16 biological species (Supplementary Data 12). The hypergeometric test is used to evaluate whether the dysregulated metabolites are enriched in one pathway[39]. The $P$ value from the hypergeometric test is also calculated for each pathway. The dysregulated pathways with $P$ values less than 0.05 are output as dysregulated pathways. The information for dysregulated pathways is output into the file named Pathway. enrichment.analysis.

**Quantitative analysis of dysregulated pathway**. MetDNA utilizes the quantitative information from the MS1 peaks to characterize the expression levels of the dysregulated pathways in a quantitative fashion. First, all the peak intensities are Pareto-scaled. If one peak group has multiple MS1 peaks, then the most abundant peak is selected to represent the quantity of the peak group. Second, if one metabolite matches multiple peaks, then the peak with the highest annotation score (Score$_{iden}$) is selected to represent the metabolite. Third, the expression level of one dysregulated pathway is calculated as the average value of all the metabolites in the pathway. The quantitative results for the metabolites and pathways are output into two files named Quantitative.pathway.metabolite.result and Quantitative. pathway.result, respectively.

**Predicting the RT**. To obtain the theoretical RTs of all the metabolites in the MRN, MetDNA utilizes the quantitative structure–retention relationship (QSRR) to construct a prediction model to generate theoretical RTs[40,41]. The RTs of metabolites under liquid chromatography (LC) highly depend on their structures

and physiochemical properties, which can be described quantitatively using molecular descriptors (MDs). With the QSRR prediction model, the input of a set of MD values for one metabolite could generate the theoretical RT. This approach requires the following data to establish a prediction model: (1) a training dataset containing a number of metabolites with experimental RTs; (2) a machine-learning-based algorithm; and (3) the MDs of metabolites. The detailed steps for the RT prediction are described below.

Step 1. Calculating the MDs. The R package rcdk [https://cran.r-project.org/web/packages/rcdk/index.html] is used to calculate the MDs of the metabolites from their SMILES structures. The SMILESs of metabolites from the in-house spectral library and the MRN were obtained using the Identifier Exchange Service of PubChem [https://pubchemdocs.ncbi.nlm.nih.gov/identifier-exchange-service]. A total of 346 MDs are calculated for each metabolite.

Step 2. Obtain a training dataset. The annotated metabolites through the spectral match are used as the training dataset to establish the prediction model. If one MS1 peak has multiple metabolite annotations, or vice versa, then the unique metabolite annotation is selected using the following criteria: (1) if one MS1 peak has multiple annotations, the one with the highest DP score is kept; (2) if one metabolite matched to multiple MS1 peaks, then the one with the highest intensity is kept; and (3) metabolite annotations with reliable adducts, such as $[M+H]^+$, $[M+Na]^+$, and $[M+NH_4]^+$ in positive ionization and $[M-H]^-$, $[M+CH3COO]^-$, and $[M+Cl]^-$ in negative ionization are selected. For the metabolites in the training dataset, MDs with more than 50% missing values (MVs) across all metabolites are removed. The remaining MVs in the training dataset are imputed using the KNN algorithm (with the R package impute). The MDs with the same values across all the metabolites are also removed. As a result, a training dataset is obtained, including a set of metabolites with experimental RTs, and the MDs for the metabolites are calculated.

Step 3. Optimize the combination of MDs for prediction. A machine-learning-based algorithm known as the random forest (RF) is used to develop the prediction model. First, the combination of MDs is optimized for prediction. An RF model is constructed using MDs as independent variables and RTs as dependent variables (with the R package randomForest). Approximately 68% of the metabolites are randomly selected as the internal training dataset, and the remaining metabolites are used as the internal test dataset. An importance value is assigned to each MD to evaluate its contribution to the prediction model. The model construction is repeated 100 times. Each time, the top five ranked MDs according to the importance values are recorded. The MDs that appear > 50 times in 100 RF models are selected as the optimized MD combination. Using the datasets in our lab, we optimized two combinations of MDs for HILIC and reverse phase (RP) separations, respectively. The optimized combination of MDs for the HILIC includes XLogP, tpsaEfficiency, WTPT.5, khs.dsCH, MLogP, nAcid, nBase, and BCUTp.1l. The optimized combination of MDs for the RP phase includes XLogP, WTPT.5, WTPT.4, ALogp2, and BCUTp.1l.

Step 4. Parameter optimization. The parameters in the RF algorithm, ntree (i.e., number of trees to grow) and mtry (i.e., number of variables randomly sampled as candidates at each split) are also optimized. The two parameters are combined together to form a set of parameter combinations. The performance of each parameter combination is evaluated using the mean squared error (MSE). The parameter combination with the smallest MSE is used to construct the final prediction model.

Step 5. Retention time prediction. With the RF-based prediction model, the theoretical RTs of all the metabolites in the spectral library and MRN are obtained using their calculated MDs. The theoretical RTs are used to improve the confidence in the metabolite annotations.

**Standard MS2 spectral library**. The standard MS2 spectral library is used to annotate the initial seed metabolites in MetDNA. The curation of the library followed the protocol in our previous publication[22]. All the MS2 spectra were acquired on Sciex TripleTOF 5600 or 6600 instruments with commercial metabolite standards. For each metabolite, the targeted product ion scans were applied to acquire the MS2 spectrum with a flow injection method. The curation of the spectral library follows the instructions and protocols in a publication from the NIST to improve spectral reproducibility[42]. In brief, for each metabolite, at least 11 MS2 spectra were acquired. The cluster of MS2 spectra with high similarities (DP > 0.7) was selected to generate a consensus MS2 spectrum. MS2 spectra at different levels of collision energy (10, 20, 30, 40, and 35 ±15 eV) were acquired. The current library in MetDNA contains 841 metabolites in total, with 841 for the positive mode and 837 for the negative mode (Supplementary Data 13).

**Reagents, fruit fly culture, and sample preparation**. LC-MS grade water ($H_2O$) and methanol (MeOH) were purchased from Honeywell (Muskegon, USA). LC-MS grade acetonitrile (ACN) was purchased from Merck (Darmstadt, Germany). Ethanol was purchased from Sinopharm (Beijing, China). Ammonium hydroxide ($NH_4OH$) and ammonium acetate ($NH_4OAc$) were purchased from Sigma-Aldrich (St. Louis, USA). Metabolite chemical standards were purchased from J&K (Beijing, China), Sigma (St. Louis, USA), Carbosynth (Berkshire, UK), TCI (Tokyo, Japan) and Energy Chemical (Shanghai, China).

Wild-type male fruit flies (FlyBase ID: FBst0005905) were cultured in standard media (temperature, 25 °C; humidity, 60%; and a 12 h light and 12 h dark cycle). At

day 3 (3-day) and day 30 (30-day), 100 fruit flies were collected and divided into 10 samples (10 flies in each sample, $n = 10$ in each group, and 20 samples in total). The fruit flies were killed with 75% ethanol. The heads of the fruit flies were collected and placed into microcentrifuge tubes. The tubes were immediately frozen with liquid nitrogen and stored at −80 °C until metabolite extraction. The *Drosophila* aging samples were defrosted on ice. The *Drosophila* aging samples were then homogenized with 200 μL of $H_2O$ and 20 ceramic beads (diameter, 0.1 mm) using a homogenizer (Precellys 24, Bertin Technologies). A mixture of ACN: MeOH (1:1, v/v; 800 μL) was added to the samples, which were then vortexed for 30 s, followed by incubation in liquid nitrogen for 1 min, and then thawed on ice. This vortex–freeze–thaw cycle was repeated three times. The samples were incubated for 1 h at −20 °C for protein precipitation, followed by centrifugation at 16,200g and 4 °C for 15 min. The supernatant solution was removed and evaporated to dryness in a vacuum concentrator (Labconco, USA). A mixture of ACN:$H_2O$ (1:1, v/v; 100 μL) was then added to reconstitute the dry extracts, followed by sonication (50 Hz, 4 °C) for 10 min. The solutions were centrifuged at 16,200g and 4 °C for 5 min to precipitate the insoluble debris. Finally, the supernatant solutions were transferred to HPLC glass vials and stored at −80 °C prior to LC-MS/MS analysis.

**LC-MS/MS analysis of *Drosophila* aging samples**. The metabolomics data acquisition for *Drosophila* aging samples was performed using a UHPLC system (1290 series; Agilent Technologies, USA) coupled to a quadruple time-of-flight mass spectrometer (TripleTOF 6600, AB SCIEX, USA). A Waters ACQUITY UPLC BEH Amide column (particle size, 1.7 μm; 100 mm (length) × 2.1 mm (i.d.)) was used for the LC separation and the column temperature was kept at 25 °C. Mobile phase A was 25 mM ammonium hydroxide ($NH_4OH$) + 25 mM ammonium acetate ($NH_4OAc$) in water, and B was ACN for both the positive (ESI+) and negative (ESI−) modes. The flow rate was 0.3 mL/min and the gradient was set as follows: 0–1 min: 95% B, 1–14 min: 95% B to 65% B, 14–16 min: 65% B to 40% B, 16–18 min: 40% B, 18–18.1 min: 40% B to 95% B, and 18.1–23 min: 95% B. The injection volume was 2 μL. All the samples were randomly injected during data acquisition.

The data acquisition was operated using the information-dependent acquisition (IDA) mode. The source parameters were set as follows: ion source gas 1 (GAS1), 60 psi; ion source gas 2 (GAS2), 60 psi; curtain gas (CUR), 30 psi; temperature (TEM), 600 °C; declustering potential (DP), 60 V, or −60 V in positive or negative modes, respectively; and ion spray voltage floating (ISVF), 5500 or −4000 V in positive or negative modes, respectively. The TOF MS scan parameters were set as follows: mass range, 60–1200 Da; accumulation time, 200 ms; and dynamic background subtract, on. The product ion scan parameters were set as follows: mass range, 25–1200 Da; accumulation time, 50 ms; collision energy, 30 or −30 V in positive or negative modes, respectively; collision energy spread, 0; resolution, UNIT; charge state, 1 to 1; intensity, 100 cps; exclude isotopes within 4 Da; mass tolerance, 10 ppm; maximum number of candidate ions to monitor per cycle, 6; and exclude former target ions, for 4 s after two occurrences.

**Data processing of the *Drosophila* aging dataset**. All 20 MS raw data files (.wiff) were separately converted to mzXML format and mgf format using ProteoWizard. The detailed parameters for the data conversion are listed in Supplementary Table 6. First, the mzXML data files were grouped into two folders (named W03 and W30) and subjected to peak detection and alignment using the R package called xcms (version 1.46.0 [https://bioconductor.org/packages/3.2/bioc/html/xcms.html]). The detailed code for XCMS processing is provided in Supplementary Note 7. The key parameters were set as follows: method = "centWave"; ppm = 15; snthr = 10; peakwidth = c(5, 40); minifrac = 0.5. The generated MS1 peak table includes the mass-to-charge ratio ($m/z$), RT, peak abundances, and other information. The peak table was then modified as follows: (1) for the first 12 columns, those named name, mzmed and rtmed were kept, and the others were deleted; (2) the first three columns were renamed name, mz and rt. The generated MS1 peak tables (one for positive mode and one for negative mode) are used for the MetDNA analysis.

Second, a sample information file (.csv format) is prepared to describe the sample group information. The first column is named sample.name, while the second one is named group. Two group names (W03 and W30) are provided.

Finally, the MS1 peak table, sample information file, and MS2 data files (.mgf format) were all uploaded to our MetDNA webserver [http://metdna.zhulab.cn] for data analysis. Positive and negative datasets were processed together. The data processing parameters for MetDNA were set as follows: Ionization polarity, Both; Liquid Chromatography, HILIC; MS Instrument, Sciex TripleTOF; Collision Energy, 30; Control Group, W03; Case Group, W30; Univariate Statistics, Student's *t*-test; Species, *Drosophila melanogaster* (fruit fly); Cutoff of *P* value, 0.01; and *P* value Adjustment, Yes. The detailed code for XCMS processing and parameter settings for MetDNA is provided in Supplementary Note 7.

**Transcriptomics data for the *Drosophila* aging samples**. The RNA-seq of the *Drosophila* aging head tissues (3-day vs. 30-day, $n = 3$ for each group) was obtained from our recent study[30] and downloaded from the Gene Expression Omnibus [https://www.ncbi.nlm.nih.gov/geo/] (GEO: GSE96654). In brief, the head tissue

samples of the fruit flies were sequenced using Illumina NextSeq 550 or Hisep 2500 platforms with single end 100 bps. The sequencing reads were mapped to the reference genome dm6 with STAR2.3.0. The read counts for each gene were calculated using HTSep-0.5.4. The count files were normalized with the R package DESeq. Finally, the FlyDatabase ID for each gene was transformed to KEGG ID with the R package clusterProfiler. The genes without mapped KEGG IDs were removed from the dataset. The final transcriptomics dataset of aging fruit flies is provided in Supplementary Data 14.

**Other metabolomics datasets**. In this study, a total of 11 datasets were used to evaluate the performance of MetDNA. The detailed information of dataset #2–11 (Supplementary Tables 2 and 3) are provided in Supplementary Note 3. The patients in dataset #10 were enrolled with the written informed consents and this study was approved by the Ethics Committee of the Tumor Hospital of Harbin Medical University (Harbin, China). The patients in dataset #11 were also enrolled with the written informed consents and this study was approved by the Ethics Committee of the Shandong Tumor Hospital (Jinan, China). The animal studies were approved by Animal Ethics and Welfare Management Committee of Interdisciplinary Research Center on Biology and Chemistry, Chinese Academy of Sciences (Shanghai, China).

**Multi-omics integration of metabolomics and transcriptomics**. A correlation network between the metabolomics and transcriptomics was constructed. To construct the correlation network at the pathway level, we first obtained quantitative pathway data from the metabolomics data (see Quantitative analysis of dysregulated pathway). For the transcriptomics data, the same method was applied. In brief, genes in the same pathway were grouped, and the mean value of the genes was then taken for each pathway to represent the quantitative information of this pathway. For pathways in the metabolomics and transcriptomics data, all the pairwise Pearson correlations were calculated to construct the correlation network at the pathway level for both types of data (Student's $t$-test, $P$ values < 0.05, only the same pathways in both types of data). The correlation network at the gene and metabolite level for each pathway was also constructed using the same method with a Pearson correlation (Student's $t$-test, $P$ values < 0.01 and absolute Pearson correlation values > 0.7). The network was visualized using Cytoscape [http://www.cytoscape.org/] (version 3.2.1).

**Reporting summary**. Further information on experimental design is available in the Nature Research Reporting Summary linked to this article.

## Code availability

MetDNA was developed using a mixture of R, JavaScript, and Python and is available for non-commercial use at http://metdna.zhulab.cn/. The webserver is currently hosted on a Linux server from Alibaba Cloud [https://www.alibabacloud.com/] with 16 cores (3.2 GHz CPU) and 32 GB RAM. The web-based software tool is easy to use for common users with limited bioinformatic background, and compatible with multiple operation systems (such as Windows, Linux, and Mac OS). With this configuration, the analysis for the *Drosophila* aging datasets (both positive and negative modes, 20 samples in total, dataset #1 in Supplementary Table 2) took about 2 h. The source code of MetDNA can be found and downloaded for scientific research purpose from the github via https://github.com/ZhuMSLab/MetDNA. A help document for using MetDNA can be found at http://metdna.zhulab.cn/metdna/help. The demo data are provided to learn how to use MetDNA at http://metdna.zhulab.cn/metdna/DemoDataset.

## Data availability

All the metabolomics datasets described in our study can be downloaded from the MetDNA website [http://metdna.zhulab.cn/metdna/DatasetsDownload] (see details in Supplementary Table 2). The metabolomics datasets of *Drosophila* aging (dataset #1) can also be accessed at MetaboLights [https://www.ebi.ac.uk/metabolights/index] (Project ID: MTBLS612 for positive and MTBLS615 for negative modes, respectively). The metabolomics datasets of mouse liver tissues (dataset #2) can also be accessed at MetaboLights [https://www.ebi.ac.uk/metabolights/index] (Project ID: MTBLS601 for positive and MTBLS606 for negative modes, respectively). A reporting summary for this article is available as a Supplementary Information file. All other data supporting the findings of this study are available from the corresponding author on reasonable request.

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

## Acknowledgements
The work is financially supported by National Natural Science Foundation of China (Grant No. 21575151 to Z.-J.Z.), Thousand Youth Talents Program from Government of China (to Z.-J.Z.), and Chinese Academy of Sciences Major Facility-based Open Research Program.

## Author contributions
Z.-J.Z. and X.S. conceived the idea and designed the algorithm and software. X.S. developed the MetDNA program. X.X. developed the webserver. Y.Y. contributed to part of MetDNA code. R.W., Y.C. and X.S. performed the sample preparation, data acquisition, and data processing. Z.M. and N.L. contributed to the RNA-seq data. X.S. and R.W. performed the data analysis. Z.-J.Z., X.S., X.X. and R.W. tested and debugged the program and webserver. Z.-J.Z. and X.S. wrote the manuscript. Z.-J.Z. supervised the project.

## Additional information

**Competing interests:** The authors declare no competing interests.

