## [Peer Review File · Nature Communications]

Reviewers' comments:

Reviewer #1 (Remarks to the Author):

Large-scale metabolite identification for untargeted metabolomics using metabolic reaction network

This paper tackles one of the key problems in the use of untargeted metabolomics: the problem of identifying signals observed in mass spectrometry.

It does so via the use of a knowledge base, specifically reaction pairs extracted from KEGG. A few papers in the past have attempted to use reaction information to improve identification, but this paper is somewhat unique in that they use the reactions (coupled with the assumption that two metabolites that are linked by a single reaction will have similar spectra) to effectively 'expand' the spectral libraries.

I have the following specific comments:

- The authors perform an experiment to validate the assumption that pairs of spectra that share a reaction have similar MS2 spectra. This is important, as it is upon this assumption that the work is based. However, I wonder if the null distribution (random pairs of metabolites that aren't linked through a reaction) is too simple? The method moves outwards one step at a time and therefore key to this working is that it is somewhat possible to distinguish metabolites that differ by one reaction with those that differ by two or more. It would have been useful to see how the cosine similarities decay as a function of the reaction 'distance' between two compounds. If we see many pairs separated by two reactions that also have cosine scores above 0.5, I would imagine it might be possible to introduce quite a few false positives (especially in areas of the network space in which we see re-configurations into different structural isomers). I would also imagine that the difference in precursor MZ for the randomly chosen samples would be (on average) larger than the difference between the RPs. Did you consider correcting for this when sampling the random RPs?

- [not really a criticism but something that might be interesting] - in the GNPS molecular networking workflow, a modified cosine score is used that allows for MS2 peaks to be shifted by the difference in precursor MZ between the two spectra. The assumption is that this means that spectra that differ by a single modification will have high scores. Seems that this would be an excellent scoring method to use in the current work?

- A recent paper in PLoS Comp Bio is perhaps also of note (<https://journals.plos.org/ploscompbiol/article?id=10.1371/journal.pcbi.1006089>) as I think there are some parallels with the proposed approach in terms of trying to "expand the net" of identification.

- It's maybe worth being careful about the use of words like "identification" - this has quite a precise definition within the metabolomics community (identification requires validation with a chemical standard) which these annotations(?) wouldn't meet. That's not to say they aren't useful, just have to be careful with terms that mean different things in different communities.

- On p6, it is stated that 18,320 MS1 peaks were extracted -- did all of these have MS2 spectra?

- p6 again - why were only 108 of the 150 metabolites selected at the second stage? The initial selection seems to be those that were identified and were in KEGG. These 150 have to all be in KEGG, no?

- p7 - I think the use of 'identified' here is too strong. Just because a match was found between a spectrum in the dataset and something one step away from something else (which is also quite putative) doesn't mean (IMO) that the spectra has been identified. I would prefer to say that XXX spectra can be putatively annotated (or similar).

- p7 - 6 metabolites were validated. What about the other 22?

- p9 - how were the 200 chemical standards chosen? Would we expect them all to belong to RPs in this data?

- p9 - I'm not sure I completely understand how the %age of identifications is computed. I think it's "how many of the 200 appear with a >0.5 cosine similarity of their true compound". If so, this tells us something about the sensitivity of the approach. However, it doesn't tell us anything about the specificity? E.g. how many other hits were there for these spectra (from traditional DBs, or the 'expanded' DBs?). This comment carries throughout all the experiments: looking at the top hits is useful, but in reality it's useful to know how many false positives do we see at a particular threshold.

- throughout - the paper could use a fairly thorough proof read to pick up grammatical errors of which there are a few.

Reviewer #2 (Remarks to the Author):

Authors proposed the propagation technique integrating the ideas of molecular-spectrum networking (GNPS) and genome-wide network annotation (BioCAN) studies (1,2,3). The developed method can be contributed to further annotations of unknown metabolites in LC-MS based metabolomics, and authors submitted many supplementary materials to validate the methodology especially for estimating the error rate of metabolite annotations. However, I found several concerns to the manuscript sentences, methodologies, and results that authors showed, and therefore I cannot recommend the current manuscript for the publication.

Major concerns:

1. Overall, the term 'identification' must be used carefully as described in the MSI (metabolomics standard initiative) report (4). This methodology does not identify metabolites, but 'effectively suggests' the metabolite candidates for unknown peaks using a practical chemical space, i.e. KEGG with the partial spectrum similarity. Namely, the approach provides level 3 'characterization' according to the MSI report. Only at the last sentence, authors stated 'MetDNA cannot exceed the limit of mass spectrometry or replace the chemical standards that ultimately confirm the unambiguous identification of metabolites.'. However, most of the manuscript sentences using the terms 'identified/identification' overestimate the MetDNA result, which can give an erroneous impression for journal readers.

2. The MetDNA accuracy highly depends on the size of chemical spaces to be used for searching structures. Authors used KEGG compound/pathway, which is the most popular database I believe in biology, to narrow down the metabolite candidates, but the KEGG pathway does not cover the large chemical space of metabolites to be monitored in living organisms compared to the other databases (5). How many metabolite candidates are listed as 'average' or 'median' for each precursor ion? For example, if the average count of metabolite candidates for each node is around 3-4, the accuracy becomes very high (80%-90%) if the definition of 'correct' annotation is set to 'within top 3' that authors do. Therefore, the accuracy should also be evaluated by several pathway databases such as SMPDB, MetaCyc, and Reactome. About this, in principle, the MetDNA method can utilize the retention times predicted by QSRR-RF approach, and therefore, the accuracy for RT predictions is very important because the program cannot distinguish isomers by the MS/MS matching algorithm while in silico fragmentation tools such as CFM-ID and CSI:FingerID can distinguish these isomers by their machine learning techniques. However, the accuracy of RT prediction was not clarified in the main manuscript although authors performed the optimization of RT tolerances.

3. As a methodology report, authors should carefully discuss the alternative programs which can be used to expand the coverage of metabolite annotations for unknown molecules. In addition, authors should benchmark the performance to alternative programs. Ultimately, the metabolic reaction network (MRN) approach can be replaced to in silico prediction tools such as CFM-ID, MetFrag, MS-FINDER, CSI:FingerID, MAGMa, and ChemDistiller with the use of KEGG metabolites as the chemical space for annotating unknown molecules. More importantly, the concept of authors approach is similar to GNPS-NAP [2] (and BioCAN [3]), and therefore, the methodological difference, novelty, and performance should be compared and discussed. I share your introduction sentence "Yet, the overall performance of such analysis is still limited by the size of standard spectral library". However, as authors also said, in the last sentence of main manuscript, "the MetDNA approach does not exceed the limitation". Nevertheless, authors contradicted GNPS and others without any suitable comparisons. Moreover, authors showed the result of SIRIUS very simply with no parameter details. Even if the KEGG database was set to the chemical space in SIRIUS, the comparison is not fair because 18,431 metabolites are registered in KEGG while authors only used 7,639 compounds. Note that the chemical space for searching structures is highly reflected to the prediction accuracy [6]. In addition, there is a concern to the method of MS/MS similarity calculation. Authors used dot-product score for the similarity calculation of MS/MS derived from different m/z ions. However, the dot product only considers the information of product ions for diagnostics. Namely, in order to seek

the metabolites sharing the same substructures with the targeted parent metabolite, the neutral loss information is very important for the similarity calculation. For example, GNPS invited the Bonanza score methodology using both product ion- and neutral loss values [7]. In addition, Dr. Stephen Stein group recently proposed new algorithm to compare the MS/MS spectra from different m/z precursor ions which also used the information of product ion and neutral loss [8].

4. The challenging task in LC-MS based metabolomics is to integrate 'multiple' ions from a same metabolite. As long as electrospray ionization (ESI) is used, one metabolite will be detected as various ionization forms including isotopic ions, different adduct forms, dimers, (sometimes) odd electron and in source fragment ions. Of these, a challenging and important task is to find the in source fragment ions. For example, 'tryptamine' is generated from 'tryptophan' in the ion source with the conventional ESI source, and therefore, the disadvantage of MS/MS matching based annotation is to provide many false positives as described in the previous report [9]. Overall, annotations can be executed easily by using several annotation tools including authors approach. However, the important thing is to estimate and exclude the false positive candidates [10]. I could not find any discussion for the estimation of false discovery rate in the list of 2000 annotated metabolites.

5. I imagine that there is substantial manual effort to obtain high quality results in this kind of networking approach [2]. For example, there is a sentence in the manuscript 'Specifically, 150 of 654 metabolites were identified by matching the calculated m/z, theoretical retention times'. About this, how did you identify the 150 metabolites? Completely automatic? Or manually checked from the candidate list that the MetDNA automatically generated? As the methodology paper, authors should clarify the scalability and the practical aspects.

Other concerns

1. About Fig.1b: Why 0.5 is used while 0.8 is used for the network construction?
2. Your definition of 'correct' annotation was set to 'within top 3 candidates' but it's much better to clarify the accuracy for top 1 candidate in the main text.
3. About low propagation of misidentified metabolites: authors evaluated the effect of miss-seeds to the network. I know that the 'density', defined as the count of edges connected to a node, is the important factor for the propagations. For example, if a replaced precursor metabolite is linked to many product metabolites, the propagation is very accelerated while the replaced precursor metabolite is only connected to one metabolite, the mis-propagation effect becomes small. Therefore, the node/edge densities for replaced metabolites should be discussed.
4. How many metabolites are newly identified by this methodology? And what did you discover new insights in biology? The 'propagation (about 2,000 metabolites can cumulatively be identified from one experiment.)' is always easy as an informatics researcher.

- Ref 2. da Silva RR et al. PLoS Comput Biol 14(4): e1006089, 2018
- Ref 3. Nicholas Alden et al. Anal. Chem., 2017, 89 (24), pp 13097–13104
- Ref 4. Lloyd W. Sumner et al. Metabolomics. 2007 Sep; 3(3): 211–221.
- Ref 5. Timothy Jewison et al. Nucleic Acids Res. 2014 Jan 1; 42(Database issue): D478–D484
- Ref 6. Ivana Blaženović et al. Journal of cheminformatics 9, 32, 2017
- Ref 7. Falkner, J. A. et al. J. Proteome Res. 7, 4614–4622 (2008).
- Ref 8. Arun S. Moorthy et al. Anal. Chem., 2017, 89 (24), pp 13261–13268
- Ref 9. Mahieu, N. G. & Patti, G. J. Anal. Chem. 89, 10397–10406 (2017).
- Ref 10. Kerstin Scheubert et al. Nature Communications 8: 1494 (2017)

Reviewer #3 (Remarks to the Author):

Shen et al., present the work 'Large-scale Metabolite Identification for Untargeted Metabolomics Using Metabolic Reaction Network' proposing the algorithm metabolic reaction network (MRN)-based recursive algorithm (MetDNA), which the authors claim 'substantially expands metabolite identifications'. The work is well written, most of the analysis are carefully described and the authors carried several validation studies to test the proposed algorithm. I believe that the approach is interesting and it is potentially useful for a large community of mass spectrometry based metabolomics users. However, I would suggest that the authors provide clarifications in several points on the text in order to assure the utility of the tool and avoid confusion for the readers.

My recommendation to the editor is for major alterations to the manuscript (but all seem addressable in reasonable time frame) as it needs many clarifications and then a re-review.

I question of this is really large. What is your frame of reference? MS-Dial was shown to analyze 40,000 mass spec data sets, dereplicator+ processed 250,000 MS files. If you are not processing 10,000s of files, it may be appropriate to remove Large-scale

My first major but easily fixed criticism concerns the indiscriminate use of the term 'identification'. While the authors acknowledged the Metabolomics the Standard Initiative (MS1) and for the validation studies clearly state

"Manual comparison with chemical standards demonstrated that 113 and 137 metabolites were detected by MS in positive and negative modes, respectively, which were considered as the Level 1 identification according to MSI21"

The authors also use the term several times, in the abstract:

'... about 2,000 metabolites can cumulatively be identified from one experiment.'

The metabolites 'identified' (I would suggest the term annotation), should be clearly stated in the text as putatively annotated metabolites (levels 2 or 3), as, as far as I understood, there were no standards acquired in the same experimental conditions for all those putative annotations or multiple chromatography methods or other orthogonal method was used.

Below I would like to describe a list of points I would advise to be clarified in the revision:

The first concept I would like to have addressed is the use of Reaction Pair (RPs) from KEGG. My first questions are: Is KEGG useful as a general database for metabolomics? Are the RPs representative of the metabolism? For example, how many metabolites of the HMDB can be mapped to RPs? Why not use the expansion of biochemical reaction concept expanded previously (<https://jcheminf.biomedcentral.com/articles/10.1186/s13321-015-0087-1>, <https://www.ncbi.nlm.nih.gov/pubmed/29176591>, <https://doi.org/10.1093/bioinformatics/bty864>)?

The authors state 'In our in-house library, more than 55.3% of reaction-paired neighbor metabolites have a DP score larger than 0.5.' and later suggest that the iterative nature of the algorithm and the scoring would correct for the 'propagation of misidentified metabolites'. Could the authors attempt to graph why ~45% of the reaction scores have DP smaller than 0.5? Does that mean that all these pairs that occur in the metabolism will not be captured by the algorithm in experimental data?

It seems that for most of the validations the author have used a small number of standards

- 113 and 137 metabolites – with (78.8%) and (82.5%) accuracy for standard spike in mouse samples;
- 107 seed metabolites from Drosophila aging dataset, for that:

‘As a result, 90.9% of metabolites could be successfully identified (Fig. 3c). The percentages for the correct, isomeric, and erroneous identifications among top 3 candidates were 81.1%, 6.3% and 12.6%.’

I’m confused, were 90% of the metabolites correctly identified or 81% were in the top 3?

Also, for NIST and MassBank

‘As noted, there are 2,608 and 1,464 metabolites in NIST and MassBank acquired on Q-TOF instruments, respectively. A total of 279 and 206” - accuracies (55.9%) and (62.1%)

- 279 and 206 where the remaining compounds after selection as described on Supplementary note 4?

For me, the authors have made a likely unintentional misleading statement due to lack of complete understanding of the literature in relation to the method introduced here (will explain below why it is misleading) at the introduction saying:

‘CSI:FingerID, ... given the high chemical diversity of metabolites, the accuracy for this approach still requires a substantial improvement.’

However, CSI:FingerID, has equal or better annotation accuracy, with a much larger validation set, searching databases much more representative than KEGG. In addition, similar tools for large scale annotations published recently, have much larger validation sets (see <https://www.nature.com/articles/s41467-018-06082-8> with more than 5k spectra). Would the author say the validation is representative for a general metabolomics experiment? What are the chemical classes represented in the validation datasets?

Another general criticism is how the combined result of the isotope, adduct, and retention time prediction fit in the general workflow. While the authors emphasize the recursive metabolic pairing, the role of these predictions is not clear in context. Should they be part of the schematic representation of the tool in Figure 1 and the result of the validation stress the product of all steps?

The other part that seems misplaced is the pathway enrichment analysis. The author presented this work at the last metabolomics conference and there was a whole section dedicated to pathway enrichment, where the use of Hypergeometric test for pathway enrichment was criticized (see

general criticism here for example - <https://www.nature.com/articles/s41598-017-15231-w>). Can the authors justify the model choice and how it fits to the whole workflow?

My last remark is for the software section:

'Software availability. MetDNA was developed using a mixture of R, JavaScript, and Python and is available for non-commercial use at <http://metdna.zhulab.cn/>. The webserver is currently hosted on a Linux server from Alibaba Cloud (<https://www.alibabacloud.com/>) with 8 cores (3.2 GHz CPUs) and 16 GB RAM. A help document for using MetDNA can be found at <http://metdna.zhulab.cn/help>. The demo data is provided to learn how to use MetDNA at <http://metdna.zhulab.cn/demo>.'

The authors really think that the software will have wide use and serve the community running in a server similar to an ordinary desktop computer? What was the running time for the analysis presented at the paper? Will the code be available? The problem when the good is not available is that in a few weeks your desktop will be dead, the student who developed the tool has moved on and the entire tool will be inaccessible to the community.

Also for the demo, I would like to see how to have access to the list of top five candidates for each ion.

Looking forward to reading the revised manuscript.

Response to the reviewers:

The authors would like to thank the reviewers for the helpful comments. We feel these comments have strengthened the manuscript considerably.

Reviewer #1:

General comment: *“This paper tackles one of the key problems in the use of untargeted metabolomics: the problem of identifying signals observed in mass spectrometry. It does so via the use of a knowledge base, specifically reaction pairs extracted from KEGG. A few papers in the past have attempted to use reaction information to improve identification, but this paper is somewhat unique in that they use the reactions (coupled with the assumption that two metabolites that are linked by a single reaction will have similar spectra) to effectively 'expand' the spectral libraries.”*

Ans: Thanks a lot for the reviewer's comment.

Comment #1: *“The authors perform an experiment to validate the assumption that pairs of spectra that share a reaction have similar MS2 spectra. This is important, as it is upon this assumption that the work is based. However, I wonder if the null distribution (random pairs of metabolites that aren't linked through a reaction) is too simple? The method moves outwards one step at a time and therefore key to this working is that it is somewhat possible to distinguish metabolites that differ by one reaction with those that differ by two or more. It would have been useful to see how the cosine similarities decay as a function of the reaction 'distance' between two compounds. If we see many pairs separated by two reactions that also have cosine scores above 0.5, I would imagine it might be possible to introduce quite a few false positives (especially in areas of the network space in which we see re-configurations into different structural isomers). I would also imagine that the difference in precursor MZ for the randomly chosen samples would be (on average) larger than the difference between the RPs. Did you consider correcting for this when sampling the random RPs?”*

Ans: We agree with the reviewer's comment and revised our method to construct the non-reaction pairs (non-RPs) in our revised manuscript (see **Supplementary Methods, Supplementary Fig. 1a and Supplementary Note 2**).

Specifically, for each metabolite, its neighbor metabolites in library (in-house, NIST17 or METLIN) were retrieved and utilized to construct the reaction pairs (RPs). Meanwhile, the metabolites in the library and have the smallest m/z error with the true neighbor metabolites were also retrieved and utilized to construct non-RPs. For example, if 'metabolite a' has a neighbor metabolite 'b' in the library, so 'a-b' is constructed as a RP. And then the 'metabolite c' with the smallest m/z error with 'metabolite b' in the library is used to construct the non-RP. Through this revision, the difference in precursor m/z for the non-RPs should be similar to the difference between the RPs. With the new constructed non-RPs, we calculated the distribution of their spectral similarity. As shown in **Fig. 1b** in the revised manuscript, the percentages of non-RPs with DP > 0.5 are 5.2%, 3.6% and 3.2% for in-house, NIST and METLIN library, respectively, which have similar results compared to those in our original manuscript (4.1%, 1.8% and 1.6% for in-house, NIST and METLIN library, respectively). The updated results successfully supported the assumption of our work.

To demonstrate how the cosine similarities decay as a function of the reaction 'distance' between two compounds, we constructed RPs and non-RPs with 2, 3, 4 and 5 steps, respectively. As shown in **Supplementary Fig. 1b**, the percentages of RPs with DP > 0.5 significantly decreased to 26.5%, 15.5%, 10.3% and 8.0% from 55.3% (with 1 step) for 2, 3, 4 and 5 reaction steps, respectively. All the results and discussion have been added in the revised manuscript and supplementary information.

Supplementary Figure 1. Construction of non-reaction pairs and the comparison of dot product, bonanza and hybrid similarity search (HSS) scores.

(a) Illustration of the construction of reaction pairs (RPs) and non-reaction pairs (non-RPs). (b) Spectral similarity between metabolites in RPs and non-RPs with different steps of metabolic reactions. (c) Comparison of dot product and bonanza scores for metabolites in RPs obtained from our in-house library. (d) Comparison of dot product and HSS scores for metabolites in RPs obtained from in-house library. (e) The percentage of reactions pairs with score > 0.5 utilizing DP, bonanza and HSS scores.

Copy of Figure 1b. Spectral similarity between metabolites in reaction and non-reaction pairs, respectively.

Comment #2: “[not really a criticism but something that might be interesting] - in the GNPS molecular networking workflow, a modified cosine score is used that allows for MS2 peaks to be shifted by the difference in precursor MZ between the two spectra. The assumption is that this means that spectra that differ by a single modification will have high scores. Seems that this would be an excellent scoring method to use in the current work?”

Ans: Thanks a lot for the reviewer’s comment. The modified cosine score used in GNPS is the bonanza score by Falkner *et al.* (*J. Proteome Res.*, 2008, **7**, 4614-4622). Another modified cosine score, namely, hybrid similarity search score (HSS score) developed by Moorthy *et al.* (*Anal. Chem.*, 2017, **89**, 13261-13268), allows for MS2 peaks to shift by the difference in precursor *m/z* between two spectra.

In the revised manuscript, we designed an experiment described as follows to compare the bonanza score, HSS score and dot product (DP) score. In brief, RPs retrieved from our in-house library were utilized to calculate DP, bonanza and HSS scores, respectively (see **Supplementary Note 1**). As shown in **Supplementary Fig. 1c**, most of bonanza scores were smaller than dot products, especially for the RPs with high DP scores. However, the DP and bonanza scores from ~74.0% of RPs had no significant differences (absolute difference < 0.2). In addition, the percentage of RPs with bonanza scores larger than 0.5 decreased to 30.9% compared to 55.3% using DP. Meanwhile, the percentage of non-RPs with bonanza scores larger than 0.5 decreased to 2.0% compared with 5.2% using DP score (**Supplementary Fig. 1e**).

On the contrary, most of HSS scores were larger than DP scores, especially for the RPs with low DP scores (**Supplementary Fig. 1d**). However, the DP and HSS scores from ~80.0% of RPs had no significant differences (absolute difference < 0.2). In addition, the percentage of RPs with HSS scores larger than 0.5 increased to 63.9% compared with 55.3% using DP score. Meanwhile, the percentage of Non-RPs with HSS scores larger than 0.5 also increased to 10.2% compared to 5.2% using DP score (**Supplementary Fig. 1e**).

Since bonanza score is a more strict scoring system than DP score. Presumably, the use of bonanza score may decrease the false positives in MetDNA, but it may also increase the false negatives and decrease the number of annotated metabolites in the same time. On the contrary, the use of HSS score may have an inverse effect (i.e., lower false negatives but higher false positive rate). Therefore, we think dot product score is appropriate enough in the MetDNA at present. If one would incorporate the bonanza or HSS score into MetDNA, the systematic optimization of score cut-off and evaluation of the validation results are required. The results and discussion have been added in our revised manuscript and supplementary information.

Comment #3: “A recent paper in *PLoS Comp Bio* is perhaps also of note (<https://journals.plos.org/ploscompbiol/article?id=10.1371/journal.pcbi.1006089>) as I think there are some parallels with the proposed approach in terms of trying to “expand the net” of identification.”

Ans: Thanks a lot for the reviewer’s comment. The core concept of GNPS-NAP tool (Network Annotation Propagation) was to connect experimental metabolic features in a molecular network according to their MS2 spectral similarity. Then, the authors hypothesized that the neighbor features with similar MS2 spectra tends to share the similar chemical structures (which is true but is opposed to our MetDNA). This hypothesis was used to improve the annotation accuracy of peaks which were first annotated by *in silico* predicted MS2 spectra. In principle, the NAP work employed the spectral similarity of neighbor features (instead of reaction-paired neighbor metabolites in MetDNA) to re-rank the annotations obtained from *in silico* predicted MS2 spectra. Finally, no recursive annotation was employed in NAP which limited the effective coverage. Therefore, we think the concept of NAP is very different from our MetDNA. The information has been added into the

discussion of our revised manuscript.

Comment #4: *"It's maybe worth being careful about the use of words like "identification" - this has quite a precise definition within the metabolomics community (identification requires validation with a chemical standard) which these annotations(?) wouldn't meet. That's not to say they aren't useful, just have to be careful with terms that mean different things in different communities."*

Ans: We agree with the reviewer's comment. The word "identification" has been changed to "annotation" in our revised manuscript.

Comment #5: *"On p6, it is stated that 18,320 MS1 peaks were extracted -- did all of these have MS2 spectra?"*

Ans: Thanks a lot for the reviewer's comment. 6,428 out of 18,320 MS1 peaks have MS2 spectra in positive mode aging fly dataset. This information has been added in our revised manuscript.

Comment #6: *"p6 again - why were only 108 of the 150 metabolites selected at the second stage? The initial selection seems to be those that were identified and were in KEGG. These 150 have to all be in KEGG, no?"*

Ans: Thanks a lot for the reviewer's comment. All 150 metabolites had KEGG IDs. However, 42 metabolites were the same as seeds in the first round. Therefore, we excluded them as new seeds for next round of annotation to reduce the redundancy and computational cost in the recursive annotation. In addition, we only counted 108 new identifications of metabolites in round 1 (*i.e.*, the second red bar in **Fig. 2a**). The description of seed metabolite selection has been added in our revised manuscript.

Comment #7: *"p7 - I think the use of 'identified' here is too strong. Just because a match was found between a spectrum in the dataset and something one step away from something else (which is also quite putative) doesn't mean (IMO) that the spectra has been identified. I would prefer to say that XXX spectra can be putatively annotated (or similar)."*

Ans: We agree with the reviewer's comment. The word "identification" has been changed to "annotation" in our revised manuscript and supplementary information.

Comment #8: *"p7 - 6 metabolites were validated. What about the other 22?"*

Ans: Thanks a lot for the reviewer's comment. The other 22 metabolites have no metabolite standards in our in-house library. To validate their annotations, we first searched them in public libraries, such as NIST, METLIN, HMDB and MassBank. Then for the metabolites which still have no MS2 spectra in the online library, we utilized the *in silico* MS2 spectra from CFM-ID (Allen *et. al.*, *Metabolomics*, 2014, **11**, 98-110) for validation.

In summary, 6 additional metabolites were validated using NIST/METLIN/HMDB library (**Supplementary Fig. 11**). The *in silico* MS2 spectra for the remaining 16 metabolites were predicted using CFM-ID, and compared to the experimental data (**Supplementary Table 1**). As a result, 6 out of 16 metabolites were validated with DP scores > 0.5, and 10 out of 16 metabolites had DP scores < 0.5. The results have been added in our revised manuscript and supplementary information.

Comment #9: *"p9 - how were the 200 chemical standards chosen? Would we expect them all to belong to RPs in this data?"*

Ans: Thanks a lot for the reviewer's comment. The 200 chemical standards were chosen from our in-house library with no bias, which were described in our recent publication (Cai *et al.*, *Methods in Molecular Biology*, 2019, **1859**, 263-274; https://link.springer.com/protocol/10.1007/978-1-4939-8757-3_15). A total of 167 chemical standards (83.5%) were included in metabolic reaction network (MRN). The information has been added in our revised manuscript and supplementary information.

Comment #10: “p9 - I'm not sure I completely understand how the %age of identifications is computed. I think it's "how many of the 200 appear with a >0.5 cosine similarity of their true compound". If so, this tells us something about the sensitivity of the approach. However, it doesn't tell us anything about the specificity? E.g. how many other hits were there for these spectra (from traditional DBs, or the 'expanded' DBs?). This comment carries throughout all the experiments: looking at the top hits is useful, but in reality it's useful to know how many false positives do we see at a particular threshold.”

Ans: Thanks a lot for the reviewer's comment. We have re-edited the description of validation experiments to make it more clearly to readers. For the first validation experiment, a total of 200 chemical standards were added into mouse liver samples as spiked-in standards, and the samples were acquired data using LC-MS/MS. Manual analysis of the experimental data with the comparison to the data of chemical standards (*m/z*, RT, and MS2 spectra) demonstrated that 113 and 137 metabolites were detected and identified in positive and negative modes, respectively. The data of chemical standards was individually measured using the single chemical standard under the same LC-MS/MS condition. The match criteria were set as *m/z* error < 25 ppm, RT error < 30 s and DP > 0.8. According to MSI, the identifications were considered as the Level 1 identification.

As a comparison, the same LC-MS/MS data set was also processed using MetDNA. MetDNA successfully annotated 89 (out of 113) and 113 (out of 137) metabolites in positive and negative modes, respectively. Therefore, we calculated the annotation rates as 78.8% (89/113) and 82.5% (113/137), respectively. We further evaluated the sensitivity and specificity thorough the calculation of true positive and false positive rates. To do so, the annotations from MetDNA were further compared to the identification results from manual analysis, and divided into three categories: correct, isomeric and erroneous annotations. Thus, the comparison is not performed by comparing the “*a >0.5 cosine similarity of their true compound*”. As a result, we think the correct annotation rate should be considered as true positive rates, which were 77.5% and 69.0% for positive mode and negative mode, respectively. Meanwhile, the erroneous annotation rate should be considered as the false positive rates, which were 10.1% and 16.8% for positive and negative modes, respectively (**Fig. 3 and Supplementary Fig. 13**). The second and third validation experiments have the similar results (**Fig. 3 and Supplementary Figs. 14 and 15**). The above results showed that the false positive rate for MetDNA is about 10-20%. However, we agreed with the reviewers that the results were obtained with the KEGG DB, and the expanded DB would give a different result. The information has been added in our revised manuscript and supplementary information.

Comment #11: “throughout - the paper could use a fairly thorough proof read to pick up grammatical errors of which there are a few.”

Ans: Thanks a lot for the reviewer's comment. We have checked and re-edited our languages for our revised manuscript and supplementary files.

Reviewer #2:

Comment #1: “Overall, the term ‘identification’ must be used carefully as described in the MSI (metabolomics standard initiative) report (4). This methodology does not identify metabolites, but ‘effectively suggests’ the metabolite candidates for unknown peaks using a practical chemical space, i.e. KEGG with the partial spectrum similarity. Namely, the approach provides level 3 ‘characterization’ according to the MSI report. Only at the last sentence, authors stated ‘MetDNA cannot exceed the limit of mass spectrometry or replace the chemical standards that ultimately confirm the unambiguous identification of metabolites.’. However, most of the manuscript sentences using the terms ‘identified/identification’ overestimate the MetDNA result, which can give an erroneous impression for journal readers.”

Ans: We agree with the reviewer’s comment. The word “*identification*” has been changed to “*annotation*” in our revised manuscript and supplementary information.

Comment #2: “The MetDNA accuracy highly depends on the size of chemical spaces to be used for searching structures. Authors used KEGG compound/pathway, which is the most popular database I believe in biology, to narrow down the metabolite candidates, but the KEGG pathway does not cover the large chemical space of metabolites to be monitored in living organisms compared to the other databases (5). How many metabolite candidates are listed as ‘average’ or ‘median’ for each precursor ion? For example, if the average count of metabolite candidates for each node is around 3-4, the accuracy becomes very high (80%-90%) if the definition of ‘correct’ annotation is set to ‘within top 3’ that authors do. Therefore, the accuracy should also be evaluated by several pathway databases such as SMPDB, MetaCyc, and Reactome. About this, in principle, the MetDNA method can utilize the retention times predicted by QSRR-RF approach, and therefore, the accuracy for RT predictions is very important because the program cannot distinguish isomers by the MS/MS matching algorithm while *in silico* fragmentation tools such as CFM-ID and CSI:FingerID can distinguish these isomers by their machine learning techniques. However, the accuracy of RT prediction was not clarified in the main manuscript although authors performed the optimization of RT tolerances.”

Ans: Thanks a lot for the reviewer’s comment. We agree with the reviewer that KEGG is the most important and popular database in biology. Therefore, we select the KEGG to demonstrate the MetDNA workflow instead of other databases. A total of 7,639 metabolites from KEGG were included into MRN, which is a relatively small number compared to other databases. **But we think it is large enough to demonstrate the improved identification accuracy provided by MetDNA.**

Do to so, we designed an experiment described as follows. The 917 dysregulated peaks (FDR corrected P -value < 0.01) in *Drosophila* aging datasets were chosen for annotation using KEGG database (7,639 metabolites; **Supporting Figure 1a for Reviewer** in next page). When only m/z match was performed (m/z error < 25 ppm), the average number of annotations was **35.3** per metabolic peak. This number means that one peak had about 35 metabolite candidates on average, which was far larger than 3-4 as the reviewer suggested. Meanwhile, the average number of annotations reduced to **16.6** per metabolic peak when both m/z and theoretical RT matches were performed (m/z error < 25 ppm; RT error < 30%), which was still much larger than 3-4. As a comparison, the average number of annotations obtained from MetDNA was only **1.6** per metabolite peaks. Although we allowed to output top 5 candidates for each peak, the average number of 1.6 indicated that most peak had only 1 or 2 candidates. **These results proved that KEGG is large enough to demonstrate the improved identification accuracy provided by MetDNA.**

Finally, we also agreed with the reviewer’s comment that other databases such as SMPDB, MetaCyc, and Reactome can be used to further increase the coverage and performance of MetDNA. We will consider

incorporating these databases in the near future.

For RT prediction, many previous studies have revealed that the prediction of retention time is not highly accurate especially for HILIC separation (Cao *et al.*, *Metabolomics*, 2015, **11**, 696-706; Domingo-Almenara *et al.*, *Anal. Chem.*, 2018, **90**, 480-489; Creek *et al.*, *Anal. Chem.*, 2011, **83**, 8703-8710). We also observed the similar large variation between experimental and predicted RTs obtained from a total of 623 metabolite standards (median relative error: 16.7%, **Supporting Figure 1b**). Therefore, we set a relative large threshold for RT match. To optimize the threshold of RT match, we first used chemical standards to confirm the annotations of 129 metabolites through m/z , RT and MS2 spectral match (*i.e.*, level 1 of identification according to MSI). The set of metabolites were used as the validation dataset. Then we compared the final identification results using predicted RT for match with the validation dataset (**Supplementary Fig. 7a**). The result showed that when the RT match threshold is set too small, the true positive rate gets lower. When the RT threshold was set as 30%, the true positive rate was 83.5%. This result supported the validity of our setting of RT match threshold. As we discussed in the previous paragraph, the addition of RT match (using the predicted RT values) effectively reduced the average number of annotations from 35.3 to 16.6 per metabolic peak.

The results and discussion have been added in our revised manuscript. More details have been provided in the revision.

Supporting Figure 1 for Reviewer.

(a) Average number of annotations using m/z match only, m/z + RT match, and MetDNA, respectively. (b) Comparison of measured and predicted RTs for 623 metabolite standards.

Comment #3: “As a methodology report, authors should carefully discuss the alternative programs which can be used to expand the coverage of metabolite annotations for unknown molecules. In addition, authors should benchmark the performance to alternative programs. Ultimately, the metabolic reaction network (MRN) approach can be replaced to *in silico* prediction tools such as CFM-ID, MetFrag, MS-FINDER, CSI:FingerID, MAGMa, and ChemDistiller with the use of KEGG metabolites as the chemical space for annotating unknown molecules. More importantly, the concept of authors approach is similar to GNPS-NAP [2] (and BioCAN [3]), and therefore, the methodological difference, novelty, and performance should be compared and discussed. I share your introduction sentence “Yet, the overall performance of such analysis is still limited by the size of

standard spectral library”. However, as authors also said, in the last sentence of main manuscript, “the MetDNA approach does not exceed the limitation”. Nevertheless, authors contradicted GNPS and others without any suitable comparisons. Moreover, authors showed the result of SIRIUS very simply with no parameter details. Even if the KEGG database was set to the chemical space in SIRIUS, the comparison is not fair because 18,431 metabolites are registered in KEGG while authors only used 7,639 compounds. Note that the chemical space for searching structures is highly reflected to the prediction accuracy [6].”

Ans: Thanks a lot for the reviewer’s comment. In revised manuscript, we performed the benchmark comparison using *in silico* prediction tool CFM-ID (Allen *et. al.*, *Metabolomics*, 2014, **11**, 98-110; see **Supplementary Note 4** for experimental details). We predicted all the MS2 spectra of metabolites in MRN using CFM-ID to construct an *in silico* MS2 spectra library, so the chemical space of this *in silico* MS2 spectral library is the same as MetDNA. Then, we utilized the *in silico* MS2 spectra library for metabolite annotation of aging fly datasets. As shown in **Fig. 3g**, 337 and 383 metabolites were annotated for positive and negative modes, respectively, and 651 metabolites were annotated in total. From the aspect of the annotation number, MetDNA provided a much higher coverage of annotation than CFM-ID with the same chemical search space.

We also used our validation experiment #1 to evaluate the annotation accuracy using CFM-ID. For the positive mode, the top 1-3 correct annotation rates for *in silico* MS2 spectral match were 57.4%, 67.6% and 70.6%, respectively. As a comparison, the correct annotation rates for MetDNA were 66.3%, 74.2% and 77.5% respectively (**Fig. 3h**). All these results demonstrated that MetDNA annotated much more metabolites and had higher annotation accuracy than *in silico* prediction tool CFM-ID. The results and discussion have been added in our revised manuscript.

Copy of Figure 3g and 3h. (g) The numbers of annotated metabolites using in-house library, CFM-ID and MetDNA in *Drosophila* aging datasets. (h) Comparison of annotation correct rate between CFM-ID and MetDNA in the same chemical search space.

We think our MetDNA algorithm is fundamentally different with GNPS-NAP and BioCAN. The core concept of GNPS-NAP tool (Network Annotation Propagation) was to connect experimental metabolic features in a molecular network according to their MS2 spectral similarity. Then, the authors hypothesized that the neighbor features with similar MS2 spectra tends to share the similar chemical structures (which is true but is opposed to our MetDNA). This hypothesis was used to improve the annotation accuracy of peaks which

were first annotated by *in silico* predicted MS2 spectra. In principle, the NAP work employed the spectral similarity of neighbor features (instead of reaction-paired neighbor metabolites in MetDNA) to re-rank the annotations obtained from *in silico* predicted MS2 spectra. Finally, no recursive annotation was employed in NAP which limited the effective coverage.

For BioCAN work, it first constructed a network using reaction information. Then the metabolites in the network were mapped to the MS1 peak table using the *m/z* match, and generated a small size network (with 452 metabolite nodes and 676 reaction edges in the paper). Then, the experimental MS2 spectra for matched features were further matched to different MS2 spectral databases (NIST, METLIN and HMDB) and *in silico* MS2 prediction tools (MetFrag and CFM-ID) to calculate node score. Then, the authors hypothesized that one annotated node with more reliable neighbor metabolites is more accurate. So the annotation score was calculated to sum all node scores from the metabolites and their neighbor metabolites. **Similar to the GNPS-NAP, BioCAN intended to re-rank the annotations obtained from the MS2 spectral databases and *in silico* MS2 prediction tools. However, no spectral similarity between reaction-paired neighbor metabolites was utilized. Similar to GNPS-NAP, there was also no recursive annotation in BioCAN.** Therefore, we think the concepts of NAP and BioCAN are very different from our MetDNA.

Finally, although there are 18,431 metabolites are registered in KEGG, only 7,639 metabolites are defined in reaction pairs by KEGG. Therefore, we only used 7,639 metabolites to construct MRN for MetDNA. We agreed to the reviewer's comment that "*the chemical space for searching structures is highly reflected to the prediction accuracy*". In Sirius, however, we cannot manually set the use of the specific 7,639 metabolites as the search space. Therefore, we removed the comparison between MetDNA and Sirius in the revised manuscript. Instead, in revised manuscript, we used the *in silico* prediction tool CFM-ID as the comparison.

In revised manuscript, we have also re-edited our descriptions about GNPS and other tools to make it more accurate. All the results and discussion have been added in our revised manuscript and supplementary information.

Comment #4: "*In addition, there is a concern to the method of MS/MS similarity calculation. Authors used dot-product score for the similarity calculation of MS/MS derived from different *m/z* ions. However, the dot product only considers the information of product ions for diagnostics. Namely, in order to seek the metabolites sharing the same substructures with the targeted parent metabolite, the neutral loss information is very important for the similarity calculation. For example, GNPS invited the Bonanza score methodology using both product ion- and neutral loss values [7]. In addition, Dr. Stephen Stein group recently proposed new algorithm to compare the MS/MS spectra from different *m/z* precursor ions which also used the information of product ion and neutral loss [8].*"

Ans: Thanks a lot for the reviewer's comment. Please refer to the reply to **Comment #2 from Reviewer 1**.

In the revised manuscript, we designed an experiment described as follows to compare the bonanza score (Falkner *et al.*, *J. Proteome Res.*, 2008, **7**, 4614-4622), Hybrid Similarity Search (HSS) score (Moorthy *et al.*, *Anal. Chem.*, 2017, **89**, 13261-13268) and dot product (DP) score. In brief, RPs retrieved from our in-house library were utilized to calculate DP, bonanza and HSS scores, respectively (see Supplementary Methods). As shown in **Supplementary Fig. 1c**, most of bonanza scores were smaller than dot products, especially for the RPs with high DP scores. However, the DP and bonanza scores from ~74.0% of RPs had no significant differences (absolute difference < 0.2). In addition, the percentage of RPs with bonanza scores larger than 0.5 decreased to 30.9% compared to 55.3% using DP. Meanwhile, the percentage of non-RPs with bonanza scores larger than 0.5 decreased to 2.0% compared with 5.2% using DP score (**Supplementary Fig. 1e**). On the

contrary, most of HSS scores were larger than DP scores, especially for the RPs with low DP scores (**Supplementary Fig. 1d**). However, the DP and HSS scores from ~80.0% of RPs had no significant differences (absolute difference < 0.2). In addition, the percentage of RPs with HSS scores larger than 0.5 increased to 63.9% compared with 55.3% using DP score. Meanwhile, the percentage of Non-RPs with HSS scores larger than 0.5 also increased to 10.2% compared to 5.2% using DP score (**Supplementary Fig. 1e**).

Since bonanza score is a more strict scoring system than DP score. Presumably, the use of bonanza score may decrease the false positives in MetDNA, but it may also increase the false negatives and decrease the number of annotated metabolites in the same time. On the contrary, the use of HSS score may have an inverse effect (i.e., lower false negatives but higher false positive rate). Therefore, we think dot product score is appropriate enough in the MetDNA at present. If one would incorporate the bonanza or HSS score into MetDNA, the systematic optimization of score cut-off and evaluation of the validation results are required. The results and discussion have been added in our revised manuscript and supplementary information.

Comment #5: *“The challenging task in LC-MS based metabolomics is to integrate ‘multiple’ ions from a same metabolite. As long as electrospray ionization (ESI) is used, one metabolite will be detected as various ionization forms including isotopic ions, different adduct forms, dimers, (sometimes) odd electron and in source fragment ions. Of these, a challenging and important task is to find the in source fragment ions. For example, ‘tryptamine’ is generated from ‘tryptophan’ in the ion source with the conventional ESI source, and therefore, the disadvantage of MS/MS matching based annotation is to provide many false positives as described in the previous report [9]. Overall, annotations can be executed easily by using several annotation tools including authors approach. However, the important thing is to estimate and exclude the false positive candidates [10]. I could not find any discussion for the estimation of false discovery rate in the list of 2000 annotated metabolites.”*

Ans: Thanks a lot for the reviewer’s comment. In MetDNA, the annotation of isotope and adduct peaks are also included. We have re-edited the part of annotation of isotope and adduct peaks to make it more clearly for readers. First, the isotope peaks of seed metabolites are annotated by MetDNA. Specially, for each seed metabolite, the program calculates the theoretical m/z and the relative intensities of isotopes peaks from the formula using the binomial and McLaurin expansion (R package “*Rdisop*”), only four isotopes peaks ([M] to [M+4]) are calculated. Then the generated isotopes peaks of seed metabolites are used to match all the MS1 peaks in the MS1 peak table according to the m/z , RT and relative intensity. The default tolerances for the m/z , RT and relative intensity are set as ± 25 ppm, ± 3 s, and $\pm 500\%$, respectively. The large tolerance for the relative intensity match is due to the inaccuracy of the experimental intensity ratios for the isotope peaks. Second, MetDNA calculates the possible m/z values for the adduct peaks of seed metabolites. The adduct table for different LC separations (HILIC or RP) and polarities (positive and negative) are listed in **Supplementary Table 13**. All the possible adduct peaks of the seed metabolites are then matched to the MS1 peak table according to their m/z and RT. The default tolerances for the m/z and RT are ± 25 ppm and ± 3 s, respectively. However, the annotation of in-source fragmentation is not included in MetDNA. But some in-source fragmentations can be well differentiated using the RT. For example, RTs for tryptamine and tryptophan are 325.1 s and 446.7 s, respectively, which generate a RT error of 37.2%. In MetDNA, true tryptamine and tryptamine from the in-source fragmentation can be well differentiated using the RTs.

In fact, in our validation experiments, we think the correct annotation rate should be considered as true positive rates, which were 77.5% and 69.0% for positive mode and negative modes in the first validation experiment, respectively. Meanwhile, the erroneous annotation rate should be considered as the false positive rates, which were 10.1% and 16.8% for positive and negative modes, respectively (**Fig. 3 and**

Supplementary Fig. 13). The second and third validation experiments have the similar results (**Fig. 3 and Supplementary Figs. 14 and 15**). The above results showed that the false positive rate for MetDNA is about 10-20%. In addition, **please refer to the reply to Comment #10 from Reviewer 1 for more discussions.** All of the discussion and results have been added in our revised manuscript and supplementary information.

Comment #6: “I imagine that there is substantial manual effort to obtain high quality results in this kind of networking approach [2]. For example, there is a sentence in the manuscript ‘Specifically, 150 of 654 metabolites were identified by matching the calculated m/z , theoretical retention times’. About this, how did you identify the 150 metabolites? Completely automatic? Or manually checked from the candidate list that the MetDNA automatically generated? As the methodology paper, authors should clarify the scalability and the practical aspects.”

Ans: Thanks a lot for the reviewer’s comment. All the data processing within MetDNA is automatic and no manual analysis is involved. The reviewer can have a try with our web server (<http://metdna.zhulab.cn/>). Specific to the reviewer’s comment, the metabolite identification in each round is also completely automatic.

We have re-edited related description to make it more clearly for readers. Briefly, first, for each round, seed metabolites were mapped to MRN to retrieve their neighbor metabolites. The MS2 spectra from the seed metabolites were assigned to their corresponding neighbor metabolites as surrogate MS2 spectra. The precursor m/z was calculated for the neighbor metabolites according to the possible adduct (such as $[M+H]^+$ or $[M+Na]^+$) in the LC condition and ionization polarity. In addition, RTs for the neighbor metabolites were also predicted. For each neighbor metabolite, the calculated m/z , predicted RT and surrogate MS2 spectrum were matched to the experimental m/z (default tolerance, 25 ppm), RT (default tolerance, 30%), and MS2 spectrum (default score cutoff, 0.5) for annotation purpose (**Supporting Figure 2 for Reviewer**). For each annotation, annotation score was also calculated (see **Online Methods**). For example, in the aging fly positive mode dataset, 654 neighbor metabolites were retrieved in the first round, and then these 654 neighbor metabolites were automatically matched to all peaks in the dataset using the criteria described above. Finally, 150 neighbor metabolites had a match with the experimental metabolic peaks. Again, the processing step is completely automatic.

Supporting Figure 2 for Reviewer.

Illustration of the annotation of neighbor metabolites in MetDNA.

Comment #7: “About Fig.1b: Why 0.5 is used while 0.8 is used for the network construction?”

Ans: Thanks a lot for the reviewer’s comment. The MRN used in the MetDNA is knowledge-based. When the

spectral similarity score cutoff is set as 0.5, the percentage of reaction pairs with DPs larger than 0.5 is 55.3%, while the percentage of non-reaction pairs with DPs larger than 0.5 is 5.2%. In contrast, when the cutoff is set as 0.8, the percentage of non-reaction pairs with DPs larger than 0.8 decreases to 2.0%. However, the percentage of reaction pairs with DPs larger than 0.8 significantly decreases to 37.3%. So we think the cut-off score of 0.5 is appropriate. The result has been added in **revised Supplementary Note 1**.

Comment #8: “Your definition of ‘correct’ annotation was set to ‘within top 3 candidates’ but it’s much better to clarify the accuracy for top 1 candidate in the main text.”

Ans: We agree to the reviewer’s comment. The accuracies for top 1 candidate are 66.3%, 61.1% and 63.6% for the first, second and third validation experiment, respectively. The results have been added in our revised manuscript and **Fig. 3**.

Comment #9: “About low propagation of misidentified metabolites: authors evaluated the effect of miss-seeds to the network. I know that the ‘density’, defined as the count of edges connected to a node, is the important factor for the propagations. For example, if a replaced precursor metabolite is linked to many product metabolites, the propagation is very accelerated while the replaced precursor metabolite is only connected to one metabolite, the mis-propagation effect becomes small. Therefore, the node/edge densities for replaced metabolites should be discussed.”

Ans: We agree to the reviewer’s comment. In the revised manuscript, we demonstrated that the mean number of edges (*i.e.*, “edge/node density”) for correct seeds, type 1, type 2, type 3 and type 4 misannotated seeds (mean \pm SEM) were 8.2 ± 2.8 , 10.8 ± 2.9 , 10.3 ± 3.9 , 7.5 ± 3.6 and 6.1 ± 2.7 , respectively (**Supporting Figure 3 for Reviewer**). We concluded that the mean number of edges had no significant differences between correct seeds and four types of misannotated seeds. These results also proved that the low propagation of misannotated seeds have no strong relationship with edge density. The results have been added in **revised Supplementary Table 11**, and the discussion has been added in our revised manuscript.

Supporting Figure 3 for Reviewer.

Edge/node densities for correct seeds, type 1, type 2, type 3 and type 4 misannotated seeds, respectively.

Comment #10: “How many metabolites are newly identified by this methodology? And what did you discover new insights in biology? The ‘propagation (about 2,000 metabolites can cumulatively be identified from one

experiment.)' is always easy as an informatics researcher."

Ans: Thanks a lot for the reviewer's comment. For the aging fly datasets, 1,180 and 1,246 metabolites were newly annotated for positive and negative modes, respectively (**Supporting Figure 4s for Reviewer**). In addition, 15 new pathways were enriched using the annotation results from MetDNA compared to those obtained from the MS2 spectral match (**Supporting Figure 4b for Reviewer**). One prominent example was the glycolysis and gluconeogenesis pathway, which was decreased with age, but increased in PRC2 deficient animals, a long-lived mutant reported in our recent study (Ma *et al.*, *eLife*, 2018, 7: e35368). In addition, MetDNA analysis revealed that arginine biosynthesis pathway had a late-onset increase in old flies, but the extent of increase was diminished in age-matched PRC2 mutants (**Fig. 5e**). Since the urea cycle included in arginine biosynthesis is the only metabolic route by which toxic ammonia can be converted into urea to be excreted from the body (Rabinovich *et al.*, *Nature*, 2015, 527, 379-383), this data suggested a negative correlation between arginine biosynthesis and urea cycle with the fly lifespan.

Finally, we compared the numbers of annotated metabolites using MetDNA and *in silico* MS2 spectral library generated from CFM-ID. Both had the same coverage for metabolite search space (*i.e.*, 7,639 metabolites). We annotated the aging fly datasets using the *in silico* MS2 spectral library from CFM-ID. The match tolerance for *m/z*, RT and DP were set as ± 25 ppm, $\pm 30\%$ and 0.5. Finally, 337 and 383 metabolites were annotated for positive and negative modes, respectively, and 651 metabolites were annotated in total (**Supporting Figure 4 for Reviewer**). This result demonstrated that MetDNA can annotate more metabolites than *in silico* MS2 spectral match with the same size of chemical space. The results and discussion have been added in our revised manuscript and supplementary information. Therefore, we could not agree to the comment "The 'propagation (about 2,000 metabolites can cumulatively be identified from one experiment.)' is always easy as an informatics researcher."

Supporting Figure 4 for Reviewer.

(a) The annotated metabolites numbers using in-house library, CFM-ID and MetDNA in *Drosophila* aging datasets. (b) Venn diagram shows the overlap of enriched pathways using MetDNA and in-house library annotated metabolites, respectively.

Reviewer #3:

General Comment: “Shen *et al.*, present the work ‘Large-scale Metabolite Identification for Untargeted Metabolomics Using Metabolic Reaction Network’ proposing the algorithm metabolic reaction network (MRN)-based recursive algorithm (MetDNA), which the authors claim ‘substantially expands metabolite identifications’. The work is well written, most of the analysis are carefully described and the authors carried several validation studies to test the proposed algorithm. I believe that the approach is interesting and it is potentially useful for a large community of mass spectrometry based metabolomics users. However, I would suggest that the authors provide clarifications in several points on the text in order to assure the utility of the tool and avoid confusion for the readers.”

Ans: Thanks a lot for the reviewer’s comment.

Comment #1: “I question of this is really large. What is your frame of reference? MS/Dial was shown to analyze 40,000 mass spec data sets, dereplicator+ processed 250,000 MS files. If you are not processing 10,000s of files, it may be appropriate to remove Large-scale.”

Ans: We agree with the reviewer’s comment. The manuscript title has been changed to “Metabolic Reaction Network-based Recursive Metabolite Annotation for Untargeted Metabolomics” in our revision.

Comment #2: “My first major but easily fixed criticism concerns the indiscriminate use of the term ‘identification’. While the authors acknowledged the Metabolomics the Standard Initiative (MSI) and for the validation studies clearly state. “Manual comparison with chemical standards demonstrated that 113 and 137 metabolites were detected by MS in positive and negative modes, respectively, which were considered as the Level 1 identification according to MSI21”

The authors also use the term several times, in the abstract:

‘... about 2,000 metabolites can cumulatively be identified from one experiment.’

The metabolites ‘identified’ (I would suggest the term annotation), should be clearly stated in the text as putatively annotated metabolites (levels 2 or 3), as, as far as I understood, there were no standards acquired in the same experimental conditions for all those putative annotations or multiple chromatography methods or other orthogonal method was used.”

Ans: We agree with the reviewer’s comment. The word “identification” has been changed to “annotation” in our revised manuscript.

Comment #3: “The first concept I would like to have addressed is the use of Reaction Pair (RPs) from KEGG. My first questions are: Is KEGG useful as a general database for metabolomics? Are the RPs representative of the metabolism? For example, how many metabolites of the HMDB can be mapped to RPs? Why not use the expansion of biochemical reaction concept expanded previously (<https://jcheminf.biomedcentral.com/articles/10.1186/s13321-015-0087-1>, <https://www.ncbi.nlm.nih.gov/pubmed/29176591>, <https://doi.org/10.1093/bioinformatics/bty864>)?”

Ans: Thanks a lot for the reviewer’s comment. KEGG is the most important and popular database in biology, and is one of the most curated database for metabolomics. In KEGG, the substrate–product metabolite pairs in a metabolic reaction were manually curated and validated for each reaction, which makes them highly reliable (Kanehisa *et al.*, *Nucleic Acids Res.*, 2014, **42**, 199-205). As a methodology article, we must first demonstrate the feasibility of the method. Therefore, we chose the KEGG database for MetDNA due to its reliable

definition of reaction pairs and neighbor metabolites. In the future, other expanded biochemical reaction databases such SMPDB, MetaCyc, and Reactome could also be considered to further expand the coverage.

Comment #4: “The authors state ‘In our in-house library, more than 55.3% of reaction-paired neighbor metabolites have a DP score larger than 0.5.’ and later suggest that the iterative nature of the algorithm and the scoring would correct for the ‘propagation of misidentified metabolites’. Could the authors attempt to graph why ~45% of the reaction scores have DP smaller than 0.5? Does that mean that all these pairs that occur in the metabolism will not be captured by the algorithm in experimental data?”

Ans: We agree with the reviewer’s comment that not all of the neighbor metabolites in MRN have the similar MS2 spectra. In original manuscript, we have partially discussed the point in the first paragraph of DISCUSSION. Briefly, there are many occasions where a single metabolic biotransformation is sufficient to significantly alter the MS2 spectra between reaction-paired neighbor metabolites. We think this is the reason that ~45% of the reaction pairs have DP scores smaller than 0.5. However, the algorithm in MetDNA does **NOT** require all neighbor metabolites share similarity in their MS2 spectra. For example, L-Arginine has 26 metabolite neighbors through different reactions. Only neighbor metabolites having similar MS2 spectra with L-Arginine could be potentially identified by MetDNA. Those metabolites without having similar MS2 spectra with L-Arginine could not be identified by MetDNA. The presence of neighbors without similar MS2 spectra with L-Arginine does not affect the identification other spectrally similar neighbors.

In addition, one metabolite may have many neighbor metabolites in different metabolic reactions. MetDNA does not require all neighbor metabolites have similar MS2 spectra with the node metabolite. If one of many neighbor metabolites has the similar MS2 spectrum as the node metabolite, it is enough to annotate this node metabolite. So even ~45% of the reaction-paired metabolites have DP score smaller than 0.5, the fact does not mean that all these metabolites could not be captured by MetDNA. The principle is simple but effective. This result and discussion have been added in our revised manuscript.

Comment #5: “It seems that for most of the validations the author have used a small number of standards - 113 and 137 metabolites – with (78.8%) and (82.5%) accuracy for standard spike in mouse samples; - 107 seed metabolites from *Drosophila* aging dataset, for that: ‘As a result, 90.9% of metabolites could be successfully identified (Fig. 3c). The percentages for the correct, isomeric, and erroneous identifications among top 3 candidates were 81.1%, 6.3% and 12.6%.’ I’m confused, were 90% of the metabolites correctly identified or 81% were in the top 3? Also, for NIST and MassBank ‘As noted, there are 2,608 and 1,464 metabolites in NIST and MassBank acquired on Q-TOF instruments, respectively. A total of 279 and 206” - accuracies (55.9%) and (62.1%) - 279 and 206 where the remaining compounds after selection as described on Supplementary note 4?”

Ans: Thanks a lot for the reviewer’s comment. MetDNA is a bioinformatic tool to annotate metabolites in real biological samples, instead of generating the predicted *in silico* MS2 spectra for annotation. In another word, MetDNA must annotate metabolites in the true biological samples. Therefore, in our validation experiments, we used (1) 200 chemical standards spiked into the biological sample, (2) in-house standard library (841 metabolites), (3) NIST and MassBank experimental libraries (2,608 and 1,464 metabolites, respectively), and (4) CFM-ID *in silico* spectral library (7639 metabolites from MRN, newly added in the revision) to perform the validations. Each validation experiment represented different confidence levels and different sizes of chemical search space. Since we cannot directly compare the MetDNA with other spectral libraries, we only

compared the annotation results from the same biological samples using different tools (**Figure 3** in the revised manuscript).

Second, it seems that the reviewer was confused about the “*annotation coverage*” and “*accuracy*” for validation experiments. In the revised manuscript, we have re-edited this portion to improve the readability. Briefly, for the second validation experiment, the annotation coverage is 90.9%, which means that 90.9% of metabolites can be annotated by MetDNA. Among the annotated metabolites, 81.1% of annotated metabolites are correct annotations from MetDNA.

For the third validation experiment, 279 and 206 metabolites were annotated using NIST and MassBank libraries. The match conditions were set as: m/z error < 25 ppm, and MS2 spectral similarity > 0.9. The annotation coverage rates are 55.9% and 62.1% for positive mode and negative mode, respectively, which indicated that 55.9% (156/279) and 62.1% (128/206) metabolites were also annotated by MetDNA. The two percentages were not “*accuracies*”. As stated in manuscript, the percentage for the correct annotation among top 3 candidates was 74.0%, which could be considered as the accuracy (or true positive rate). Again, it seems that the reviewer was confused about the annotation coverage and accuracy for validation experiments.

In addition, we found that 59 out of 279 for positive mode and 54 out of 206 metabolites for negative mode annotated by NIST/MassBank were not included in MRN. This explained why MetDNA had relative lower annotation coverages in validation experiment #3. In addition, although there were 2,608 and 1,464 metabolites in NIST and MassBank, but only 10-15% of metabolites were annotated in real biological samples. The low ratio may be due to the insufficient measurement of metabolite in libraries on the particular LC-MS condition, or uncharacterized spectral variations across different LC-MS instruments and laboratories (*i.e.*, between the experimental data and library data). These revisions have been added into the revised manuscript.

Comment #6: “For me, the authors have made a likely unintentional misleading statement due to lack of complete understanding of the literature in relation to the method introduced here (will explain below why it is misleading) at the introduction saying:

‘CSI:FingerID, ... given the high chemical diversity of metabolites, the accuracy for this approach still requires a substantial improvement.’

However, CSI:FingerID, has equal or better annotation accuracy, with a much larger validation set, searching databases much more representative than KEGG. In addition, similar tools for large scale annotations published recently, have much larger validation sets (see <https://www.nature.com/articles/s41467-018-06082-8> with more than 5k spectra). Would the author say the validation is representative for a general metabolomics experiment? What are the chemical classes represented in the validation datasets?”

Ans: Thanks a lot for the reviewer’s comment. In revised manuscript, we have re-edited our descriptions about CSI:FingerID and other tools to make it more accurate.

As stated in the reply to **Comment #5**, MetDNA is a bioinformatic tool to annotate metabolites in real biological samples, instead of generating the predicted *in silico* MS2 spectra for annotation. MetDNA must annotate metabolites in the true biological samples. This is why we cannot directly validate MetDNA using other large-scale MS2 spectral libraries including NIST, MassBank, and *in silico* MS2 spectral prediction tools, such as CSI:FingerID and CFM-ID. Therefore, in our validation experiments, we used (1) 200 chemical standards spiked into the biological sample, (2) in-house standard library (841 metabolites), (3) NIST and MassBank experimental libraries (2,608 and 1,464 metabolites, respectively), and (4) CFM-ID *in silico* spectral library (7639 metabolites from MRN, newly added in the revision) to perform the validations (**Figure 3** in the revised manuscript). Each validation experiment represented different confidence levels and different

sizes of chemical search space. Therefore, we think our validation experiments represented a general metabolomics experiment.

For the comparison to CFM-ID, please refer to **Figure 3g and 3h**, and the reply to **Comment #3 from Reviewer 2** for the details. In brief, the results demonstrated that MetDNA annotated much more metabolites and had higher annotation accuracy than *in silico* prediction tool CFM-ID when using the same chemical search space. However, we also agreed that the incorporation of other expanded biochemical reaction databases such SMPDB, MetaCyc, and Reactome could further expand the coverage of MetDNA (see the reply to **Comment #3**). The results and discussion have been added in our revised manuscript.

Comment #7: *“Another general criticism is how the combined result of the isotope, adduct, and retention time prediction fit in the general workflow. While the authors emphasize the recursive metabolic pairing, the role of these predictions is not clear in context. Should they be part of the schematic representation of the tool in Figure 1 and the result of the validation stress the product of all steps?”*

Ans: Thanks a lot for the reviewer’s comment. In **revised Supplementary Fig. 2**, we have added the detailed workflow for isotope, adduct and retention time prediction. We have re-edited the manuscript to make it more clearly for readers.

In brief, retention prediction was performed after the MS2 spectral matching in MetDNA. Please refer to **Online Methods** for details (page 25-26 in manuscript file). Isotope and adduct peak annotation was performed after the annotation of reaction-paired neighbor metabolites. Please refer to **Online Methods** for details (page 22-23 in manuscript file).

Comment #8: *“The other part that seems misplaced is the pathway enrichment analysis. The author presented this work at the last metabolomics conference and there was a whole section dedicated to pathway enrichment, where the use of Hypergeometric test for pathway enrichment was criticized (see general criticism here for example - <https://www.nature.com/articles/s41598-017-15231-w>). Can the authors justify the model choice and how it fits to the whole workflow?”*

Ans: We agreed with the point that when calculating the *P*-values using the hypergeometric test or the Fisher test in pathway enrichment analysis, the significance levels depend on the size of background database (Barupal *et al.*, *Sci. Rep.*, 2017, 7: 14567). In MetDNA, both MRN and pathway information were obtained from the KEGG database, so the background database is static and constant for the pathway enrichment analysis. In addition, as a general tool for metabolomics, we have to incorporate a common and popular pathway analysis tool for the users of MetDNA. Therefore, we chose the common hypergeometric test for pathway enrichment.

Meanwhile, we compared the performances of the hypergeometric test and Kolmogorov-Smirnov (KS) test for the pathway enrichment analysis. The results showed that all the enriched pathways from hypergeometric test were included in the enriched pathways from the KS test (**Supporting Figure 5 for Reviewer**). We will consider to integrate Kolmogorov-Smirnov (KS) test as one option for pathway enrichment in MetDNA in the future.

Supporting Figure 5 for Reviewer.

Venn diagram demonstrated the overlap of the enriched pathways using KS test and hypergeometric test obtained from the aging fly dataset, respectively.

Comment #9: “My last remark is for the software section:

‘Software availability. MetDNA was developed using a mixture of R, JavaScript, and Python and is available for non-commercial use at <http://metdna.zhulab.cn/>. The webserver is currently hosted on a Linux server from Alibaba Cloud (<https://www.alibabacloud.com/>) with 8 cores (3.2 GHz CPUs) and 16 GB RAM. A help document for using MetDNA can be found at <http://metdna.zhulab.cn/help>. The demo data is provided to learn how to use MetDNA at <http://metdna.zhulab.cn/demo.’>

The authors really think that the software will have wide use and serve the community running in a server similar to an ordinary desktop computer? What was the running time for the analysis presented at the paper? Will the code be available? The problem when the good is not available is that in a few weeks your desktop will be dead, the student who developed the tool has moved on and the entire tool will be inaccessible to the community.

Also for the demo, I would like to see how to have access to the list of top five candidates for each ion.

Looking forward to reading the revised manuscript.”

Ans: Thanks a lot for the reviewer’s comment. To ensure the computational power of MetDNA, we have updated the Linux server with 16 cores (3.2 GHz CPU) and 32 GB RAM provided by Alibaba Cloud (<https://www.alibabacloud.com>). The web-based software tool is easy to use for common users with limited bioinformatic background, and compatible with multiple operation systems (such as Windows, Linux and Mac OS). With this configuration, the analysis for the aging fly dataset (both positive and negative modes, 20 samples in total, dataset #1 in **Supplementary Table 2**) took about two hours. The source code of MetDNA is available upon request for scientific research purpose. In our group, we have two staff scientists and two graduate students are responsible to maintain the MetDNA webserver. We promise to ensure the maintenance of MetDNA (including debug and upgrade) for at least two years after its publication.

The detailed annotation result of MetDNA is provided as a .csv file named as “MRN.annotation.result.csv”. Users can evaluate the identification results on their own.

Reviewers' comments:

Reviewer #1 (Remarks to the Author):

I thank the authors for their detailed response to my comments, and the additional experimentation they have done to reinforce their points.

I have a query about a couple of responses though. In response to mine (and another reviewer's) comments about the use of the dot product, the authors did a comparison between the dot product and the bonanza score. However, as far as I can tell, the Bonanza score is subtly different from the score used in GNPS (which is described in <https://www.pnas.org/content/109/26/E1743>). The Bonanza score is a dot product between two extended peak lists, where each list is extended with the neutral losses. This is similar to the GNPS system but the actual scoring system is different. The GNPS system finds the optimal score -- i.e. each peak can either be matched as itself, or its corresponding loss (but not both) and the choice is made via a greedy algorithm. As such, the standard dot product falls out as a special case and therefore, GNPS scores should always be equal to, or greater than the dot product (losses are only used if they give a higher score than a normal match). This is clearly different from the Bonanza score (the authors' results demonstrate this: their Bonanza scores are invariably *lower* than the DPs).

So, I don't think the comparison with the bonanza score is really relevant as (to the best of my knowledge) the Bonanza score isn't used within Metabolomics. If a comparison is to be provided, I think it ought to be with the actual GNPS scoring system.

Secondly, I comment the authors on performing the additional RP and nonRP comparisons. It is reassuring that when the null distribution is matched for precursor m/z the number of FPs does not increase substantially. However, I am still somewhat concerned about the potential for FPs afforded by the 25% of step 2 RPs that have a cosine score of >0.5. Although 25% is lower, it is presumably 25% of a much larger set (i.e. metabolites will typically have quite a few step 2 RPs. To put this figure into context, I think it's important that this is converted into absolute numbers. In other words, I guess it is more interesting to know in absolute terms how many FPs might occur by finding multi-step metabolites and assuming they are one step away rather than the proportion of things that are 2 steps away that would be FPs.

Reviewer #2 (Remarks to the Author):

Dear authors:

I read your answers for my comments and your revised manuscript. In fact, your manuscript was improved compared to the first draft. I think, as a methodology article, your article will be contributed to metabolomics community. However, I still found several concerns in the revised manuscript in addition to the answers: several answers were still unclear and not enough for me. I listed my concerns below.

(1) You mentioned in the answer for comment #2: "This number means that one peak had about 35 metabolite candidates on average". This is seriously skeptical. Or, you should explain the detail more. I checked your supplementary table "183666_1_data_set_3475414_pk8k8b.xlsx" which includes a reaction list by using 7,639 metabolites. Then, I searched all of your kegg compound ID by kegg rest api, and I tried to retrieve molecular formula information for each kegg ID. First of all, four of 7,639 (C03607, C03786, C04283, C19751) were not found (by 404 error protocol) in the current KEGG api. And even if I searched these IDs at an web browser, these IDs are no more listed in KEGG database. Furthermore, I checked the FTP server of KEGG, and these records were not included. Please explain the reason why you included these and please write the detail of when you downloaded it to make your reaction network. Second, 1200 of 7,639 were not organic molecules including metals or not the metabolites defined as a strict chemical structure (e.g. C05321:

C8H15N5O2R2, "R" is used for the abbreviation). It means that your chemical space for searching structures is not "7,639" but "6,435": if your method scopes the metal ions like Mg, please modify the above number by yourself. Thirdly, I calculated the exact mass values for all of organic formula (i.e. 6,435 chemicals). Here, I found that 109 of 6435 was out of the MS1 mass scan range (60-1200) that you used. Consequently, your chemical space should be written as now "6,326". Finally, I performed the statistics to count metabolites per a peak. For example, if I use the tolerance of 0.01Da for searching metabolites for one m/z value's peak, two metabolites are registered in between 60.02 and 60.03 in your chemical space (see Table that I uploaded). Here, I calculated the average value of metabolite counts when the mass tolerance was set to 0.01Da. Surprisingly, the average is "1.97" which is far from the number of 35 you estimated. Next, I used "25 ppm" which is exactly the same as what you used for searching metabolites. The average of metabolite count was "1.91"! I uploaded "to authurs.xlsx" for you. You also said in the answer for comment #5: "In MetDNA, the annotation of isotope and adduct peaks are also included". It means that when one node, i.e. a precursor m/z's peak is searched to the chemical space, the adduct form has been already determined: the exact mass of metabolite ion can definitively be calculated. This is seriously unclear for me. Please explain the details of how you counted the average number. See this article: <https://www.pnas.org/content/112/41/12580.long>. They carefully examined the "random" error. Moreover, why you used a relatively wider mass tolerance, i.e. 25 ppm, for searching metabolites? As long as QTOF-MS is used in metabolomics, the mass accuracy is basically less than 10 ppm.

(2) You did not perform the database evaluation by using other pathway databases even though I asked authors as "the accuracy should also be evaluated by several pathway databases such as

SMPDB, MetaCyc, and Reactome". However, you mentioned in "Discussion" section of main text as "The high correct rate of annotation is not dependent on the size of chemical spaces used in MetDNA" without any evaluations of other databases.

(3) Overall, your definitions for 'Correct' and 'Annotated' are very ambiguous. First, what is 'annotated' for you? Did you mean that one peak is defined as 'annotated' even if several candidates exceeded the score cut off? Did you mean that even if several other candidates (35 metabolites/peak) are still listed as the candidate list, you used the "annotated" term for one node? Moreover, your 'correct' definition is that the 'correct' metabolite is listed within top 3 candidates, right? Both of your definitions SERIOUSLY misleads the readers of Nature Communications. Therefore, write your definition CLEARLY in the main text NOT in Online Methods. For example, metabolomics researchers use the term "correct" when the metabolite is listed as 'top' candidate. And they use the term "annotated" if one peak is reliably assigned by a structure although it is not confirmed by authentic standard.

(4) There are several concerns in CFM-ID comparisons. First, as long as I read your supplementary note, you used the pre-trained (i.e. default) model for generating in silico spectral annotations. It's ok. However, you should state this detail in the main text. This is actually serious drawback of machine learning techniques as used in CSI:FingerID and CFM-ID which require the substantial training set for own experimental data to get a good prediction model. However, when a good prediction model is set by using the suitable training sets, these programs often provide better results than other programs (see CASMIE 2016/2017). Therefore, please clarify "what you did" in the main text as well. Also, please discuss your result with the consideration of the above things. Moreover, you mentioned in some places as "MetDNA provided a much higher coverage of annotation than CFM-ID with the same chemical search space". This sentence SERIOUSLY misleads the readers because the "coverage" of CFM-ID should be the same as that of MetDNA since you use the same chemical space, i.e. 6,326 structures. Probably, you found that the metabolite candidates exceeding YOUR cut-off were less than those of what MetDNA provided. It means that the number of annotated metabolites that YOU defined was less in CFM-ID than that of MetDNA. Therefore, you mentioned like "MetDNA provided a much higher coverage of annotation than CFM-ID". However, without the strict definition of 'annotated' as I mentioned above, this sentence is unclear. If users want to increase the 'coverage' only, what users have to do is to decrease the score-cut off, right? Therefore, Figure 3h is ok, but figure 3g confused.

(5) You mentioned in the answer for comment #3 "we think the concepts of NAP and BioCAN are very different from our MetDNA". Not VERY. You also used the idea of spectral similarity-based networking for metabolite annotations. The reason why GNPS does not use the metabolic reaction information is that they focus on natural products and they try to annotate new chemical structures, i.e. Unknown-Unknowns which have never been reported elsewhere. On the other hand, you focus on the "known" metabolic pathway defined by KEGG mainly focusing on "primary metabolisms". Therefore, somebody may mention that your method cannot be used for natural product researches and untargeted lipidomics where the metabolite pathway and enzyme selectivity are not well defined. Please discuss those backgrounds. You mentioned in the introduction "the overall performance of such analysis is still limited by the size of experimental MS spectral databases and the prediction accuracy of in silico MS2 spectra". Rather, it should be mentioned like "when the annotations are practically targeted to known chemical spaces, the information of molecular

reactions can be incorporated to increase the confidence of metabolite annotations. Therefore, we develop a strategy, MetDNA...". Also, you compared three methodologies including dot product, bonanza, and hybrid similarity search. I was surprised because you provided the dot product score is the best option for primary metabolite searching. It's ok. However, other two methods should be very useful for natural product- and lipidomics researches because very specific neutral losses to define a metabolite class can be observed and must be checked in the MS/MS spectra of natural products and lipids. Therefore, discuss the advantage/disadvantage of these three methods, and clarify the fact that the dot product scoring is suitable for connecting the nodes defined by KEGG reaction network.

(6) You mentioned that "Therefore, we could not agree to the comment "The 'propagation (about 2,000 metabolites can cumulatively be identified from one experiment.)' is always easy as an informatics researcher". See this article:

<https://pubs.acs.org/doi/abs/10.1021/acs.analchem.8b04698>. The title is "Structure annotation of all mass spectra in untargeted metabolomics". They identified/annotated (MSI level 1,2 and 3) the mass spectra by several informatics tools. These authors will be able to say the similar sentence to yours like "our methodology could annotate several thousands of metabolites". It means that only 'annotation' is very easy for informatics researchers. Even if they do not use your method, they could annotate all of MS/MS spectra in urine data. Your method and their method should have many false positive identifications, and therefore, the accuracy is very important fact to evaluate this kind of paper. Can you clearly write that your methodology is superior to Ivana et al? For example, the chemical space should be much wider in the paper because they used several structure databases which include pubchem. I do not know the accuracy aspect however, they used CSI:FingerID for MSI level 3 annotations, and the program should be superior to CFM-ID according to the CSI:FingerID's paper. Cite Ivana et al, and clarify your method advantage. Moreover, again, clarify your definitions of "correct" and "annotated".

(7) In Figure 1b, clarify how many metabolite records are included for the evaluation of mass spectral similarity in RP. I am wondering if the enough number of metabolites is evaluated or not. Your in-house library contains 841 metabolites, but how many metabolites can be evaluated as the reaction pair?

(8) Figure 3 (a-c): The term 'correct' will mislead the journal readers.

(9) You mentioned "we think the correct annotation rate should be considered as true positive rates, which were 77.5% and 69.0% for positive mode and negative mode, respectively. Meanwhile, the erroneous annotation rate should be considered as the false positive rates, which were 10.1% and 16.8% for positive and negative modes, respectively". I do not think so because your 'correct' definition is really different from the real correct definition. Again, 'correct' means that 'this peak should be this structure', and the false positive/negative rates should be evaluated by this definition. However, your 'correct' means that 'the peak may be this (candidate 1) or this (candidate 2) or this (candidate 3)'.

Reviewer #3 (Remarks to the Author):

Reviewer #3:

The authors have addressed all comments and clarifications in a rigorous manner, except comment 9.

Comment nine was not a strong response but will let the decision fall to the editor and their policies. But below I indicate my concern. Other than this issue, which is sufficient for me to reject a paper as it does not adhere to FAIR practices. However that is my bias and thus think it should be left to the editor to decide how rigorous they adhere to FAIR principles. In fact I wished the authors would rethink their response to comment 9 for the betterment of science and so that this paper becomes a role model of how science should be shared. Aside from this, I congratulate the authors on nice work and a wonderful paper.

Comment #9: "My last remark is for the software section:

'Software availability. MetDNA was developed using a mixture of R, JavaScript, and Python and is available for non-commercial use at <http://metdna.zhulab.cn/>. The webserver is currently hosted on a Linux server from Alibaba Cloud (<https://www.alibabacloud.com/>) with 8 cores (3.2 GHz CPUs) and 16 GB RAM. A help document for using MetDNA can be found at <http://metdna.zhulab.cn/help>. The demo data is provided to learn how to use MetDNA at <http://metdna.zhulab.cn/demo>.'

The authors really think that the software will have wide use and serve the community running in a server similar to an ordinary desktop computer? What was the running time for the analysis presented at the paper? Will the code be available? The problem when the good is not available is that in a few weeks your desktop will be dead, the student who developed the tool has moved on and the entire tool will be inaccessible to the community.

Also for the demo, I would like to see how to have access to the list of top five candidates for each ion.

Looking forward to reading the revised manuscript."

Ans: Thanks a lot for the reviewer's comment. To ensure the computational power of MetDNA, we have updated the Linux server with 16 cores (3.2 GHz CPU) and 32 GB RAM provided by Alibaba Cloud (https://www.alibabacloud.com). The web-based software tool is easy to use for common users with limited bioinformatic background, and compatible with multiple operation systems (such

as Windows, Linux and MacOS). With this configuration, the analysis for the aging fly dataset (both positive and negative modes, 20 samples in total, dataset #1 in Supplementary Table 2) took about two hours. The source code of MetDNA is available upon request for scientific research purpose

REVIEWER COMMENT: do the journal policies allow this?

In our group, we have two staff scientists and two graduate students are responsible to maintain the MetDNA webserver. We promise to ensure the maintenance of MetDNA (including debug and upgrade) for at least two years after its publication. The detailed annotation result of MetDNA is provided as a .csv file named as "MRN.annotation.result.csv". Users can evaluate the identification results on their own.

REVIEWER COMMENT: we should be careful with this wording of "promise". The objective answer would be, how long is our funding? There a lot of great tools that run out of support, regardless of the best effort of the authors and thus this is not a viable solution longterm. There are already too many such examples in the mass spec literature.

REVIEWER COMMENT: Finally not making your source code available (and you can even specify the license free for academics for example but license fees for the industry and makes it unusable for others. My experience over and over with "available upon request" is that labs are unable to (not due to malicious reasons rather it is usually due to not enough resources to manage the interactions and developing new tools). Not making it readily available also makes sure it is cited less frequent and it also ensures that if people want to license it from you won't as they cannot judge how they implement it. Basically, you are decreasing your own visibility and scientific efforts.

However, it is up to the editors to decide what their policies are as they are not clear. I suggest that the authors have an internal discussion based on my comments and then have a discussion with the editors if their accessibility is acceptable according to the editorial policies.

I dont need to re-review the paper at this time and again nice job on the work to the authors.

Response to the decision and Review comments

First of all, we thank the Reviewers for their comments and suggestions. Below, *italic* fonts are Reviewers' comments, followed by our response in **blue**, detailing how we have revised the manuscript and added new experiments to address the concerns from the Reviewers. We hope that you will now find the manuscript acceptable for publication in *Nature Communications*.

Reviewer #1:

General Comment: *"I thank the authors for their detailed response to my comments, and the additional experimentation they have done to reinforce their points."*

Response: We thank the Reviewer for highlighting the significance of the work and the impact to the field.

Comment #1: *"I have a query about a couple of responses though. In response to mine (and another reviewer's) comments about the use of the dot product, the authors did a comparison between the dot product and the bonanza score. However, as far as I can tell, the Bonanza score is subtly different from the score used in GNPS (which is described in <https://www.pnas.org/content/109/26/E1743>). The Bonanza score is a dot product between two extended peak lists, where each list is extended with the neutral losses. This is similar to the GNPS system but the actual scoring system is different. The GNPS system finds the optimal score -- i.e. each peak can either be matched as itself, or its corresponding loss (but not both) and the choice is made via a greedy algorithm. As such, the standard dot product falls out as a special case and therefore, GNPS scores should always be equal to, or greater than the dot product (losses are only used if they give a higher score than a normal match). This is clearly different from the Bonanza score (the authors' results demonstrate this: their Bonanza scores are invariably *lower* than the DPs). So, I don't think the comparison with the bonanza score is really relevant as (to the best of my knowledge) the Bonanza score isn't used within Metabolomics. If a comparison is to be provided, I think it ought to be with the actual GNPS scoring system."*

Copy of Supplementary Figure 1e and 1f.

(e) Comparison of dot product and GNPS scores of metabolites in RPs from in-house library. (f) Percentage of pairs with DPs > 0.5 utilizing DP, bonanza, HSS and GNPS scores in in-house library.

Response: We appreciate the issue raised by the Reviewer. Now in the new revision, we have added the comparison results using GNPS score (Watrous, *et al.*, *Proc. Natl. Acad. Sci. U. S. A.*, **2012**, 109, 1743-1752). As noted, all GNPS scores between metabolites in RPs were equal to or greater than dot product (DP) scores (**Supplementary Fig. 1e**). However, the DP and GNPS scores from ~87.0% of RPs had no significant differences (absolute difference < 0.2). The percentage of RPs with GNPS scores larger than 0.5 increased to 67.3% compared with 55.3% using DP score. Meanwhile, the percentage of Non-RPs with GNPS scores larger than 0.5 also increased to 10.4% compared to 5.2% using DP score (**Supplementary Fig. 1f**).

These results demonstrated that the GNPS score is similar to the hybrid similarity search (HSS) score developed by Moorthy *et al.* (*Anal. Chem.*, **2017**, 89, 13261-13268). It is possible that the use of GNPS and HSS scores will facilitate to increase the annotated metabolite numbers in MetDNA. However, it may also subsequently increase the false positives given the fact that the spectral similarity scores between metabolites in non-RPs become increased (5.2% using DP vs. 10.4% using GNPS). To incorporate the GNPS score into MetDNA, the systematic optimizations of score cut-off and evaluation of the validation results must be required. We have added these data in the Result (**Supplementary Fig. 1e and 1f**) and discussed this in the Discussion in the revised manuscript.

Comment #2: “Secondly, I comment the authors on performing the additional RP and nonRP comparisons. It is reassuring that when the null distribution is matched for precursor *mz* the number of FPs does not increase substantially. However, I am still somewhat concerned about the potential for FPs afforded by the 25% of step 2 RPs that have a cosine score of >0.5. Although 25% is lower, it is presumably 25% of a much larger set (*i.e.* metabolites will typically have quite a few step 2 RPs. To put this figure into context, I think it's important that this is converted into absolute numbers. In other words, I guess it is more interesting to know in absolute terms how many FPs might occur by finding multi-step metabolites and assuming they are one step away rather than the proportion of things that are 2 steps away that would be FPs.”

Response: Notably, the false spectral similarity between metabolites in non-RPs, but not in RPs, causes the false positive (FP) annotations. Since the cosine similarities between metabolites in non-RPs remained around 5.0% as shown in **Supplementary Fig. 1b**, we reason that the false positive annotation is not related to the reaction steps. We would like to emphasize the fact that this conclusion was also supported by the results in the section of “Low propagation of misannotated metabolites” (**Fig. 4c-f**). In brief, we constructed four types of misannotated metabolites compared to correct seed metabolites. MetDNA utilized these misannotated seed metabolites to perform MRN-based recursive annotation and compared with the control (*i.e.*, using correct seeds). Because the correct seeds were replaced by misannotated seeds, **those misannotated metabolites were actually connected and compared to their non-RP metabolites for MRN based recursive analysis**. In the construction of four types of misannotated metabolites, the spectral similarity between the misannotated metabolite and its false neighbor metabolite increased from type 1 to type 4 misannotated metabolites (see **Fig. 4f**). The evidence in **Fig. 4d** clearly demonstrated that with the increased spectral similarity between metabolites in non-RPs, the false positive annotations also increased from type 1 to type 4 misannotated metabolites. Together, the false spectral similarity between metabolites in non-RPs caused the false positive annotations, which is independent of reaction steps.

In addition, the spectral similarity between metabolites in RPs determined the capability and coverage to annotate the neighbor metabolites within one or multiple reaction steps. The higher similarity indicated the higher chance to annotate the neighbor metabolites. Clearly, the cosine similarities significantly decay along the increasing reaction steps between metabolites in RPs. This evidence implied that the challenge for annotation of metabolites becomes increased with multiple reaction steps. **This decayed spectral similarity is**

only related to the annotation capability and coverage instead of false positive annotations. In another word, if two reaction-paired metabolites are spectrally similar, it can be annotated by MetDNA regardless of their reaction steps (either one or multiple). In MetDNA, we set up to 3 reaction steps for the search for neighbor metabolites.

Finally, as suggested by the Reviewer, we have added a copy of **Supplementary Fig. 1b** with the absolute numbers in y-axis. In brief, among 1-step reactions, there were 300 RPs and 28 non-RPs with $DP > 0.5$. Among 2-step reactions, there were 573 RPs and 138 non-RPs with $DP > 0.5$. Among 3-step reactions, there were 708 RPs and 251 non-RPs with $DP > 0.5$. Among 4-step reactions, there were 894 RPs and 495 non-RPs with $DP > 0.5$. Among 5-step reactions, there were 1,041 RPs and 755 non-RPs with $DP > 0.5$.

Taken together, we concluded that the false spectral similarity between metabolites in non-RPs causes the false positive annotations. Meanwhile, the spectral similarity between metabolites in RPs determined the capability and coverage to annotate the neighbor metabolites within one or several reaction steps. We have discussed this in the Discussion in the revised manuscript.

Reviewer #2:

General Comment: “Dear authors:

I read your answers for my comments and your revised manuscript. In fact, your manuscript was improved compared to the first draft. I think, as a methodology article, your article will be contributed to metabolomics community. However, I still found several concerns in the revised manuscript in addition to the answers: several answers were still unclear and not enough for me. I listed my concerns below.”

Response: We appreciate the positive comments from the Reviewer very much and have worked hard to address the comments.

Comment #1: *“You mentioned in the answer for comment #2: “This number means that one peak had about 35 metabolite candidates on average”. This is seriously skeptical. Or, you should explain the detail more. I checked your supplementary table “183666_1_data_set_3475414_pk8k8b.xlsx” which includes a reaction list by using 7,639 metabolites. Then, I searched all of your kegg compound ID by kegg rest api, and I tried to retrieve molecular formula information for each kegg ID. First of all, four of 7,639 (C03607, C03786, C04283, C19751) were not found (by 404 error protocol) in the current KEGG api. And even if I searched these IDs at an web browser, these IDs are no more listed in KEGG database. Furthermore, I checked the FTP server of KEGG, and these records were not included. Please explain the reason why you included these and please write the detail of when you downloaded it to make your reaction network. Second, 1200 of 7,639 were not organic molecules including metals or not the metabolites defined as a strict chemical structure (e.g. C05321: C8H15N5O2R2, “R” is used for the abbreviation). It means that your chemical space for searching structures is not “7,639” but “6,435”: if your method scopes the metal ions like Mg, please modify the above number by yourself. Thirdly, I calculated the exact mass values for all of organic formula (i.e. 6,435 chemicals). Here, I found that 109 of 6435 was out of the MS1 mass scan range (60-1200) that you used. Consequently, your chemical space should be written as now “6,326”. Finally, I performed the statistics to count metabolites per a peak. For example, if I use the tolerance of 0.01Da for searching metabolites for one m/z value’s peak, two metabolites are registered in between 60.02 and 60.03 in your chemical space (see Table that I uploaded). Here, I calculated the average value of metabolite counts when the mass tolerance was set to 0.01Da. Surprisingly, the average is “1.97” which is far from the number of 35 you estimated. Next, I used “25 ppm” which is exactly the same as what you used for searching metabolites. The average of metabolite count was “1.91”! I uploaded “to authurs.xlsx” for you. You also said in the answer for comment #5: “In MetDNA, the annotation of isotope and adduct peaks are also included”. It means that when one node, i.e. a precursor m/z’s peak is searched to the chemical space, the adduct form has been already determined: the exact mass of metabolite ion can definitively be calculated. This is seriously unclear for me. Please explain the details of how you counted the average number. See this article: <https://www.pnas.org/content/112/41/12580.long>. They carefully examined the “random” error. Moreover, why you used a relatively wider mass tolerance, i.e. 25 ppm, for searching metabolites? As long as QTOF-MS is used in metabolomics, the mass accuracy is basically less than 10 ppm.”*

Response: We apologize for the lack of clarity. We hereby provide detailed explanation to address the comments.

First of all, the KEGG compound database in our manuscript was downloaded using an R package “KEGGREST” in Bioconductor (<https://bioconductor.org/packages/release/bioc/html/KEGGREST.html>) on March 7th, 2017. In this version of KEGG compound database, four metabolites (C03607, C03786, C04283, and C19751) were all included. We have provided this version of KEGG compound database as an R file for

the Reviewer (file named as “kegg.compound” in **Supplementary Data 1**). The database can be loaded in R environment using “load” function.

Secondly, the KEGG reaction pair database was originally defined as **RPAIR** in KEGG (Kanehisa, *et al.*, *Nucleic Acids Res.*, **2014**, 42, 199-205). This database was also obtained using the R package “KEGGREST” on March 7th, 2017. Please refer to this version of KEGG reaction pair database as an R file for the Reviewer (file named as “kegg.rpair” in **Supplementary Data 1**). The database can be loaded in R environment using “load” function. In original manuscript, we did not filter any reaction pairs from the KEGG reaction pair database, and all of reaction pairs were directly imported into R package “igraph” (<https://igraph.org/r/>) to construct the MRN. The detailed information of obtaining of KEGG databases has been added in our revised manuscript (“Online Methods” section). During revision, we have carefully re-checked the compounds in our MRN, and found that 1,200 out of 7,639 were not organic molecules or the metabolites with a strict chemical structure. Consequently, we corrected the chemical space of MRN as 6,439 metabolites in our manuscript. Indeed, 109 of 6,439 metabolites were out of MS data acquisition range (60-1200 Da) with 46 metabolites with mass < 60 Da and 63 metabolites with mass > 1200 Da, respectively. Nevertheless, these metabolites can be searched either through different adduction forms (*e.g.*, [M+K]⁺) or multiple charged forms (*e.g.*, [M+2H]²⁺), or simply using a wider MS data acquisition range. Therefore, these metabolites remain kept in the database. To better clarify this, we have revised manuscript (“Online Methods” section).

Thirdly, to better explain how we annotated peaks by *m/z* match, we calculated the average annotation number per peak described as the following. In brief, 917 dysregulated peaks (FDR corrected *P*-value < 0.01) in *Drosophila* aging datasets were chosen for annotation using MRN database (6,439 metabolites in total). For each metabolite, different adduct ions generated different *m/z* values (*e.g.*, [M+H]⁺ and [M+Na]⁺ for positive mode). The adduct ion table used in our study was provided in **Supplementary Table 13** (24 and 16 adduct ions for positive and negative modes, respectively). **Then, *m/z* value for each of 917 peaks was used to match *m/z* values of adduct ions from all metabolites in MRN database. It is important to note that the *m/z* values of adduct ions instead of accurate mass of metabolites were used.** Then, the *m/z* error was calculated, and if the *m/z* error was less than the pre-set cutoff (*i.e.*, 25 ppm), this metabolite candidate with the specific adduct form was assigned to the MS1 peak as an annotation. Finally, the average annotation number per peak was calculated using the annotation results from all of 917 peaks. We apologize for the lack of clarity and we have provided the detailed information and detailed annotation table in revised supplementary (**Supplementary Note 8** and **Supplementary Data 2**).

It is important to note that, in *m/z* match, one could not know the ion adduct for each MS1 peak before the match. Therefore, all possible adduct ions must be considered. For the comparison, we calculated the extract mass values of all metabolites in MRN, and the *m/z* values of all metabolites with adduction ions. The frequency distributions were plotted in **Supporting Fig. 1 for the Reviewer 2**. The mass tolerance window was set as 0.01 Da. Clearly, the number of metabolites within a mass tolerance window is significantly less than that with adduct ions.

Finally, different TOF MS instruments have different resolutions and mass accuracies. Specifically for Q-TOF MS in our study (Sciex TripleTOF and Agilent Q-TOF), the resolution and mass accuracy is *m/z* dependent. The low mass range (*e.g.*, 50-400 Da) has a significantly lower resolution and mass accuracy than high mass range (*e.g.*, 800-1200 Da). In addition, low ion intensity may also have a detrimental effect on mass accuracy. Therefore, it is best to use a wider mass tolerance than the theoretical tolerance of an instrument to cover all possible situations (*i.e.*, relative low-resolution TOF instrument, low mass ion, low-intensity, *etc.*). However, this tolerance can be adjusted by users. The similar suggested tolerance was also given in our previously published protocol (Zhu *et al.*, *Nature Protocols*, **2013**, 8, 451-460;

<https://www.nature.com/articles/nprot.2013.004>). We quoted the original statement in Page 455 for Reviewer's reference, "*The default and maximum tolerance of 30 ppm is generally acceptable for Q-TOF experiments; adjust the parameters as appropriate for your specific mass spectrometer. Generally, it is best to use a slightly wider window than the theoretical tolerance for an instrument.*"

Supporting Figure 1 for Reviewer.

The frequency distributions of extract masses for 6,439 metabolites and m/z values for 6,439 metabolites with adduction ions. The mass tolerance window was set as 0.01 Da.

Comment #2: "You did not perform the database evaluation by using other pathway databases even though I asked authors as "the accuracy should also be evaluated by several pathway databases such as SMPDB, MetaCyc, and Reactome". However, you mentioned in "Discussion" section of main text as "The high correct rate of annotation is not dependent on the size of chemical spaces used in MetDNA" without any evaluations of other databases."

Response: We agree with the Reviewer, and we have changed text in the Discussion now read as "The high correct rate of annotation using MetDNA can be demonstrated even with a relatively small chemical space such as KEGG". KEGG is undoubtedly an important and popular database in the field of biology, and one of the most curated databases for metabolomics. As a methodology article, we have however, in our study tried to first demonstrate the feasibility of the method, and thus, we chose the KEGG database for MetDNA due to its reliable definition of reaction pairs and neighbor metabolites. We agree that SMPDB, MetaCyc and Reactome are also very useful pathway databases. However, it will take a lot time to manually curate them, especially identifying the correct reaction pairs and remove duplications. Both the Editor(s) of *Nature Communications* and we believe the addition of other pathway databases cannot be accomplished within a reasonable revision time. Many important questions and systematic evaluations should be addressed by future developments.

Comment #3: “Overall, your definitions for ‘Correct’ and ‘Annotated’ are very ambiguous. First, what is ‘annotated’ for you? Did you mean that one peak is defined as ‘annotated’ even if several candidates exceeded the score cut off? Did you mean that even if several other candidates (35 metabolites/peak) are still listed as the candidate list, you used the “annotated” term for one node? Moreover, your ‘correct’ definition is that the ‘correct’ metabolite is listed within top 3 candidates, right? Both of your definitions **SERIOUSLY** misleads the readers of *Nature Communications*. Therefore, write your definition **CLEARLY** in the main text **NOT** in Online Methods. For example, metabolomics researchers use the term “correct” when the metabolite is listed as ‘top’ candidate. And they use the term “annotated” if one peak is reliably assigned by a structure although it is not confirmed by authentic standard.”

Response: Thank you for appreciating the significance and impact of our study.

Firstly, we have carefully investigated the definitions from Metabolomics Standards Initiative (MSI, Sumner, *et al.*, *Metabolomics*, **2007**, 3, 211-221), and assured **the annotations given by MetDNA should belong to level 3 of identification according to MSI**. The definition of level 3 of metabolite identification is given as follows: “Putatively characterized compound classes (e.g. based upon characteristic physicochemical properties of a chemical class of compounds, or by spectral similarity to known compounds of a chemical class).” In MetDNA, it first identified the initial seed metabolites using a small tandem spectral library (acquired from the chemical standards), and utilized their experimental MS2 spectra as surrogate spectra to annotate their reaction-paired neighbor metabolites. The metabolite annotations were achieved through the spectral similarity match to the seed metabolites. As a result, the annotation results given by MetDNA should be level 3 of identification according to MSI. We have added a statement in Main Text read as “According to definitions from Metabolomics Standards Initiative (MSI), the annotation results given by MetDNA (except the initial seed metabolites) are level 3 of annotation”.

In addition, Blazenovic *et al.* recently reported how they annotated all mass spectra in urine samples according to MSI definition (*Anal. Chem.*, **2019**, 91, 2155-2162). The authors defined the annotation from the bioinformatic tool CSI:FingerID as level 3 annotation. For the Reviewer, we quoted the original statement from the publication abstract, “we used the *in silico* fragmentation tool CSI:FingerID and the new NIST hybrid search to annotate all further compounds (MSI level 3)”. MetDNA is also developed as a bioinformatic tool for metabolite annotation, which is similar to CSI:FingerID.

Having that in mind, we finally clarified the annotation results given by MetDNA (except for the seed metabolites) being as MSI level 3 annotation. **Importantly, the definition of annotation is not related to the candidate numbers (or candidate number) per peak**. For example, Blazenovic *et al.* also reported multiple annotations per peak using CSI:FingerID, “Scored results of all isomeric structures were exported as CSV files. For structures returned by biodatabase searches, CSI:FingerID yielded up to 130 results per MS/MS scan and up to 10,000 structure candidates per MS/MS spectrum in PubChem queries.” In total, their analyses also outputted the **top 3 annotations** as the final annotation results in the annotation table (https://pubs.acs.org/ccindex.cn/doi/suppl/10.1021/acs.analchem.8b04698/suppl_file/ac8b04698_si_002.xlsx).

Secondly, in our manuscript, **the “correct” is defined as the correctly identified structure in the top 3 ranked output**. We have carefully edited the related statements in the manuscript to make it more clearly for readers. The definition of “correct” in our manuscript is also consistent with other metabolomics software tools. Ranking multiple metabolite annotations for one peak is a common practice for bioinformatic software in untargeted metabolomics (e.g., XCMS-Online, MetFrag, CFM-ID and CSI:FingerID and so on). For example, in CSI:FingerID, users can decide how many candidates to be remained by setting the parameter “candidate”. In their peer-reviewed and published data, the authors evaluated different methods and found that, as quoted: “We achieve 63.5% **correct identifications in the top five output**; next come FingerID with 36.1%

and CFM-ID with 36.0%.” Again, the description and definition of “correctly identified” in CSI:FingerID publication is consistent with result demonstrated in **Fig. 3** in our manuscript (**Supporting Fig. 2**). In order to make it clear to reader, we clarified that the “correct” in MetDNA manuscript is defined as the correctly identified structure in the top 3 ranked output.

Fig. 2. Methods evaluation: percentage of correctly identified structures found in the top k output of the different methods, for maximum rank $k = 1, \dots, 20$. Searching $N = 5,923$ compounds from the combined Agilent and GNPS dataset in PubChem (A) and the $N = 4,773$ biocompounds from the combined dataset in the biobdatabase (B). Identification rates 50% and 66.7% marked by dashed lines.

Supporting Figure 2 for Reviewer.

The copy of **Fig. 2** and its legend from CSI:FingerID publication (Duhrop, *et al.*, *Proc. Natl. Acad. Sci. U. S. A.*, **2015**, 112, 12580-12585)

Comment #4: “There are several concerns in CFM-ID comparisons. First, as long as I read your supplementary note, you used the pre-trained (i.e. default) model for generating *in silico* spectral annotations. It’s ok. However, you should state this detail in the main text. This is actually serious drawback of machine learning techniques as used in CSI:FingerID and CFM-ID which require the substantial training set for own experimental data to get a good prediction model. However, when a good prediction model is set by using the suitable training sets, these programs often provide better results than other programs (see CASMIE 2016/2017). Therefore, please clarify “what you did” in the main text as well. Also, please discuss your result with the consideration of the above things. Moreover, you mentioned in some places as “MetDNA provided a much higher coverage of annotation than CFM-ID with the same chemical search space”. This sentence **SERIOUSLY** misleads the readers because the “coverage” of CFM-ID should be the same as that of MetDNA since you use the same chemical space, i.e. 6,326 structures. Probably, you found that the metabolite candidates exceeding YOUR cut-off were less than those of what MetDNA provided. It means that the number of annotated metabolites that YOU defined was less in CFM-ID than that of MetDNA. Therefore, you mentioned like “MetDNA provided a much higher coverage of annotation than CFM-ID”. However, without the strict definition of ‘annotated’ as I mentioned above, this sentence is unclear. If users want to increase the ‘coverage’ only, what users have to do is to decrease the score-cut off, right? Therefore, Figure 3h is ok, but figure 3g confused.”

Response: We agree with the Reviewer, and we have re-edited the statements related to CFM-ID in the Main Text (“Validation of MetDNA analysis accuracy” section), and clarified that we used the pre-trained model in CFM-ID for generating *in silico* MS2 spectra for annotation. In the Main Text, we also added the related statement that the use of local dataset for training helps to improve the performance of machine learning techniques such as CSI:FingerID and CFM-ID. In practice, several other publications (*Proc. Natl. Acad. Sci. U. S. A.*, **2015**, 112, 12580-12585; *Anal. Chem.*, **2019**, 91, 2155-2162) also used pre-trained model for generating *in silico* MS2 spectra for annotation. Now the text is read as “We predicted all the MS2 spectra of

metabolites in MRN using CFM-ID with the pre-trained model and recommended parameters to construct an *in silico* MS2 spectra library, so the chemical space of this *in silico* MS2 spectral library is the same as MetDNA. It is worthy to note that although we and others used the pre-trained model in CFM-ID to generate *in silico* MS2 spectra, the use of local dataset for training helps to improve the performance of these machine learning based prediction tools.” (“Validation of MetDNA analysis accuracy” section).

As suggested by the Reviewer, we have changed the text now read as “MetDNA provided a much higher number of annotation metabolites than CFM-ID with the same chemical search space.” (“Validation of MetDNA analysis accuracy” section) Again, the annotations given by both MetDNA and CFM-ID are MSI level 3 annotations.

In addition, the comparison of the number of annotation metabolites of MetDNA to CFM-ID was provided in **Supplementary Note 4**. In brief, the detected metabolic peaks (datasets of validation experiment #1) were matched in *in silico* MS2 spectral library from CFM-ID. The *m/z* error cutoff was set as 25 ppm, and the retention time (RT) cutoff was set as 30%. The dot product (DP) cutoff was set as 0.3, 0.4, 0.5, 0.6, 0.7, 0.8 and 0.9, respectively. As noted, the annotation number decreased with the increasing cutoff score (**Copy of Fig. S13c**). To compare two methods with identical parameters (*i.e.*, *m/z* error, RT error and DP), we set the cutoff of DP score as 0.5. **Under the same parameters for both MetDNA and CFM-ID**, the result in **Fig. 3g** was obtained.

Copy of Figure S13c. The annotation rates for *in silico* MS2 spectral match with different DP cutoffs.

Comment #5: “You mentioned in the answer for comment #3 “we think the concepts of NAP and BioCAN are very different from our MetDNA”. Not VERY. You also used the idea of spectral similarity-based networking for metabolite annotations. The reason why GNPS does not use the metabolic reaction information is that they focus on natural products and they try to annotate new chemical structures, *i.e.* Unknown-Unknowns which have never been reported elsewhere. On the other hand, you focus on the “known” metabolic pathway defined by KEGG mainly focusing on “primary metabolisms”. Therefore, somebody may mention that your method cannot be used for natural product researches and untargeted lipidomics where the metabolite pathway and enzyme selectivity are not well defined. Please discuss those backgrounds. You mentioned in the introduction “the overall performance of such analysis is still limited by the size of experimental MS spectral databases

and the prediction accuracy of in silico MS2 spectra". Rather, it should be mentioned like "when the annotations are practically targeted to known chemical spaces, the information of molecular reactions can be incorporated to increase the confidence of metabolite annotations. Therefore, we develop a strategy , MetDNA...". Also, you compared three methodologies including dot product, bonanza, and hybrid similarity search. I was surprised because you provided the dot product score is the best option for primary metabolite searching. It's ok. However, other two methods should be very useful for natural product- and lipidomics researches because very specific neutral losses to define a metabolite class can be observed and must be checked in the MS/MS spectra of natural products and lipids. Therefore, discuss the advantage/disadvantage of these three methods, and clarify the fact that the dot product scoring is suitable for connecting the nodes defined by KEGG reaction network."

Response: We agree with Reviewer's comment that GNPS focuses on natural products and is highlighted by annotating new chemical structures. MetDNA, however, mainly focuses on the annotation of known metabolites for primary metabolisms. It is well known that KEGG database includes very limited numbers of natural products and lipids. Thus MetDNA is not applicable for natural product annotation or untargeted lipidomics where the metabolic pathways and reactions are not well defined. We have discussed this potential drawback of MetDNA in our revised manuscript ("Discussion" section).

As suggested by the Reviewer, we have changed the text now read as "When the annotations are practically targeted to known chemical spaces in primary metabolisms, the information of metabolic reactions can be incorporated to increase the confidence of metabolite annotations." ("Introduction" section)

Additionally, we have discussed four scoring methods for MS/MS spectral match (DP, bonanza, HSS and GNPS) in our revised manuscript ("Discussion" section). For MetDNA, dot product score is suitable for connecting the nodes defined in MRN. Other scores such as GNPS and HSS should be very useful to annotate new chemical structures for natural products and lipids.

Comment #6: "You mentioned that "Therefore, we could not agree to the comment "The 'propagation (about 2,000 metabolites can cumulatively be identified from one experiment.)' is always easy as an informatics researcher". See this article: <https://pubs.acs.org/doi/abs/10.1021/acs.analchem.8b04698>. The title is "Structure annotation of all mass spectra in untargeted metabolomics". They identified/annotated (MSI level 1,2 and 3) the mass spectra by several informatics tools. These authors will be able to say the similar sentence to yours like "our methodology could annotate several thousands of metabolites". It means that only 'annotation' is very easy for informatics researchers. Even if they do not use your method, they could annotate all of MS/MS spectra in urine data. Your method and their method should have many false positive identifications, and therefore, the accuracy is very important fact to evaluate this kind of paper. Can you clearly write that your methodology is superior to Ivana et al? For example, the chemical space should be much wider in the paper because they used several structure databases which include pubchem. I do not know the accuracy aspect however, they used CSI:FingerID for MSI level 3 annotations, and the program should be superior to CFM-ID according to the CSI:FingerID's paper. Cite Ivana et al, and clarify your method advantage. Moreover, again, clarify your definitions of "correct" and "annotated"."

Response: We thank the Reviewer for raising this point, and we have changed the text now read as "about 2,000 metabolites can cumulatively be annotated from one experiment with a relatively small chemical search space in MetDNA" ("Discussion" section).

As suggested by Reviewer, we have cited and discussed the recent publication (Blazenovic *et al.*, *Anal. Chem.*, **2019**, 91, 2155-2162) in our revised manuscript ("Discussion" section). In Blazenovic *et al.*'s paper,

they used multiple methods and very large chemical search spaces for metabolite annotation, including 1,102 authentic standards, MoNA and NIST17 libraries (13,808 compounds), CarniBlast library (2,400 acylcarnitine species), CSI:FingerID (a filtered version of Pubchem containing 270,000 structures) and NIST17 hybrid search). Using CSI:FingerID alone, the authors demonstrated that 728 out of 6,447 MS/MS spectra (11.3%) were assigned to chemical structures. Therefore, although about 2,000 metabolites can cumulatively be annotated from one experiment with a relatively small chemical search space in MetDNA, expanding the chemical spaces in MetDNA and combined with other tools in the future could effectively increase the number of annotated metabolites. The discussion has been added in our revised manuscript (“Discussion” section).

The definitions of “correct” and “annotated” can refer to our response to **Comment #3**. In brief, in our manuscript, the “correct” is defined as the corrected identified structure in the top 3 ranked output. We have carefully changed this in the manuscript.

Comment #7: *“In Figure 1b, clarify how many metabolite records are included for the evaluation of mass spectral similarity in RP. I am wondering if the enough number of metabolites is evaluated or not. Your in-house library contains 841 metabolites, but how many metabolites can be evaluated as the reaction pair?”*

Response: This data was provided in **Supplementary Note 1**. A total of 542, 220 and 248 reaction pairs were retrieved from our in-house spectral library, the NIST17 library and METLIN library, respectively.

Comment #8: *“Figure 3 (a-c): The term ‘correct’ will mislead the journal readers.”*

Response: Please refer to our response to **Comment #3**. In brief, in our manuscript, the “correct” is defined as the correctly identified structure in the top 3 ranked output. To make it more clearly for readers, the term “Correct” in **Fig. 3 (a-c)** have been changed to “Correct (among top 3 candidates)” in the revised **Fig. 3**.

Comment #9: *“You mentioned “we think the correct annotation rate should be considered as true positive rates, which were 77.5% and 69.0% for positive mode and negative mode, respectively. Meanwhile, the erroneous annotation rate should be considered as the false positive rates, which were 10.1% and 16.8% for positive and negative modes, respectively”. I do not think so because your ‘correct’ definition is really different from the real correct definition. Again, ‘correct’ means that ‘this peak should be this structure’, and the false positive/negative rates should be evaluated by this definition. However, your ‘correct’ means that ‘the peak may be this (candidate 1) or this (candidate 2) or this (candidate 3)’.”*

Response: Please kindly refer to our response to **Comment #3**. In brief, in our manuscript, the “correct” is defined as the correctly identified structure in the top 3 ranked output. We have carefully changed the text now read as *“Collectively, these validation experiments demonstrated that the overall performance of MetDNA attains approximately 70-80% for correct annotations among top 3 candidates and 80-90% for correct formula annotation (correct and isomer) among top 3 candidates. In addition, the correct annotation rate among top 3 candidates should be considered as true positive rate, which was 70-80% in different validation experiments. Meanwhile, the erroneous annotation among top 3 candidates rate should be considered as the false positive rate, which was 10-20% in different validation experiments”* (see “Validation of MetDNA analysis accuracy” section).

Reviewer #3:

Comment #1: *“The authors have addressed all comments and clarifications in a rigorous manner, except comment 9. Comment nine was not a strong response but will let the decision fall to the editor and their policies. But below I indicate my concern. Other than this issue, which is sufficient for me to reject a paper as it does not adhere to FAIR practices. However that is my bias and thus think it should be left to the editor to decide how rigorous they adhere to FAIR principles. In fact I wished the authors would rethink their response to comment 9 for the betterment of science and so that this paper becomes a role model of how science should be shared. Aside from this, I congratulate the authors on nice work and a wonderful paper.*

Comment #9: “My last remark is for the software section:

‘Software availability. MetDNA was developed using a mixture of R, JavaScript, and Python and is available for non-commercial use at <http://metdna.zhulab.cn/>. The webserver is currently hosted on a Linux server from Alibaba Cloud (<https://www.alibabacloud.com/>) with 8 cores (3.2 GHz CPUs) and 16 GB RAM. A help document for using MetDNA can be found at <http://metdna.zhulab.cn/help>. The demo data is provided to learn how to use MetDNA at <http://metdna.zhulab.cn/demo>.’

The authors really think that the software will have wide use and serve the community running in a server similar to an ordinary desktop computer? What was the running time for the analysis presented at the paper? Will the code be available? The problem when the good is not available is that in a few weeks your desktop will be dead, the student who developed the tool has moved on and the entire tool will be inaccessible to the community.

Also for the demo, I would like to see how to have access to the list of top five candidates for each ion.

Looking forward to reading the revised manuscript.”

Ans: Thanks a lot for the reviewer’s comment. To ensure the computational power of MetDNA, we have updated the Linux server with 16 cores (3.2 GHz CPU) and 32 GB RAM provided by Alibaba Cloud (<https://www.alibabacloud.com>). The web-based software tool is easy to use for common users with limited bioinformatic background, and compatible with multiple operation systems (such as Windows, Linux and MacOS). With this configuration, the analysis for the aging fly dataset (both positive and negative modes, 20 samples in total, dataset #1 in Supplementary Table 2) took about two hours. The source code of MetDNA is available upon request for scientific research purpose

REVIEWER COMMENT: do the journal policies allow this?

In our group, we have two staff scientists and two graduate students are responsible to maintain the MetDNA webserver. We promise to ensure the maintenance of MetDNA (including debug and upgrade) for at least two years after its publication. The detailed annotation result of MetDNA is provided as a .csv file named as “MRN.annotation.result.csv”. Users can evaluate the identification results on their own.

REVIEWER COMMENT: we should be careful with this wording of “promise”. The objective answer would be, how long is our funding? There a lot of great tools that run out of support, regardless of the best effort of the authors and thus this is not a viable solution longterm. There are already too many such examples in the mass spec literature.

REVIEWER COMMENT: Finally not making your source code available (and you can even specify the license free for academics for example but license fees for the industry and makes it unusable for others. My experience over and over with “available upon request” is that labs are unable to (not due to malicious reasons rather it is usually due to not enough resources to manage the interactions and developing new tools). Not making it readily available also makes sure it is cited less frequent and it also ensures that if people want to license it from you won’t as they cannot judge how they implement it. Basically, you are decreasing your

own visibility and scientific efforts.

However, it is up to the editors to decide what their policies are as they are not clear. I suggest that the authors have an internal discussion based on my comments and then have a discussion with the editors if their accessibility is acceptable according to the editorial policies.

I don't need to re-review the paper at this time and again nice job on the work to the authors.”

Response: Thanks a lot for the Reviewer's comment. Now we decided to provide the source code of MetDNA in github (<https://github.com/ZhuMSLab/MetDNA>). Users can access the source code with the GNU General Public License restrictions.

To ensure sustainable access to the MetDNA server, we have renewed the cloud sever service fee for MetDNA webserver until December 20th, 2023 (renewable every 5 years). Then the users will not worry about the long-term funding availability to support MetDNA running. Our commitment is to provide long-term maintenance and technical support of the MetDNA server.